# Beginning with You: Perceptual-Initialization Improves Vision–Language Representation and Alignment

## Abstract

We introduce *Perceptual-Initialization* (PI), a paradigm shift in visual representation learning that incorporates human perceptual structure *during the initialization phase* rather than as a downstream fine-tuning step. By integrating human-derived triplet embeddings from the NIGHTS dataset to initialize a CLIP vision encoder, followed by self-supervised learning on YFCC15M, our approach demonstrates significant zero-shot performance improvements—*without any task-specific fine-tuning*—across 29 zero-shot classification and two retrieval benchmarks. On ImageNet-1K, zero-shot gains emerge after approximately 15 epochs of pre-training. Benefits are observed across datasets of various scales, with improvements manifesting at different stages of the pre-training process depending on dataset characteristics. Our approach consistently enhances zero-shot Top-1 accuracy, Top-5 accuracy, and retrieval recall (e.g., R@1, R@5) across these diverse evaluation tasks, without requiring any adaptation to target domains. These findings challenge the conventional wisdom of using human-perceptual data primarily for fine-tuning and demonstrate that embedding human perceptual structure *during early representation learning* yields more capable and vision–language-aligned systems that generalize immediately to unseen tasks. Our work shows that "beginning with you"—starting with human perception—provides a stronger foundation for general-purpose vision-language intelligence.

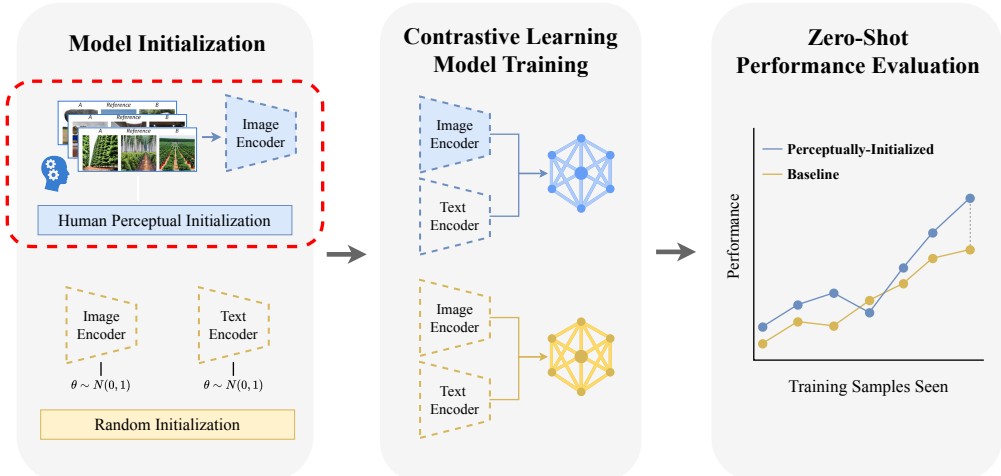

Figure 1: **Perceptual-Initialization (PI) yields faster, stronger zero-shot performance. Model initialization.** The image encoder is pre-biased with human triplet-similarity judgments from the *NIGHTS* dataset, while a control model is fully random-initialized. **Model training.** Both models are then trained with the same image–text contrastive objective on YFCC15M. **Zero-shot evaluation.** Without any task-specific fine-tuning, the perceptually-initialized model (blue) consistently outperforms the random baseline (gold).

## 1 INTRODUCTION

Deep networks are path–dependent: two models that share architecture, data, and even hyperparameters can still end up in markedly different regions of the loss landscape, exhibiting distinct internal geometries, saliency maps, and top-1 accuracies, when their weights are seeded with different random numbers (Mehrer et al., 2020; Madhyastha & Jain, 2019; Picard, 2021). Outlier ("black-swan") seeds can overshoot or undershoot the mean ImageNet score by several percentage points, a phenomenon linked to whether the initial point falls inside a favorable "Goldilocks" basin of the loss landscape (Russakovsky et al., 2015; Fort & Scherlis, 2018). Across common benchmarks, variance introduced solely by the random seed often rivals or exceeds other stochastic factors such as data shuffling(Bouthillier et al., 2021; Jordan, 2023).

At step $t = 0$, the weight matrix already defines a basis over which gradients are projected; early updates therefore amplify directions present in the initialization rather than exploring the full space uniformly. Put differently, the curvature and alignment of activation subspaces are locked in before any data are seen, channelling optimization into a restricted trajectory. If stochastic seeds can bias learning so strongly, purposeful priors injected at the same moment should exert an even greater and potentially beneficial influence.

A number of large-scale resources now characterize human perceptual similarity with some scale:

- **THINGS** contains 4.7 M pairwise similarity judgements for 1 854 everyday objects, together with low-dimensional, interpretable SPoSE embedding (Hebart et al., 2019; 2020).
- **NIGHTS** provides 20 k synthetic image triplets covering colour, pose, and semantic variations, each annotated with a two-alternative forced-choice perceptual judgement (Fu et al., 2023).

These datasets have powered a wave of post-hoc alignment methods but importantly can also be used to seed models before large-scale optimization begins (Zhang et al., 2018; Fu et al., 2023; Sundaram et al., 2024; Muttenthaler et al., 2023; 2024; Croce et al., 2025a; Zhao et al., 2025).

We initialize a Vision Transformer (ViT) trained to reproduce NIGHTS triplet embeddings, thereby infusing a human embedding into the weight space prior to any image text contrastive learning(Schroff et al., 2015; Dosovitskiy et al., 2021; Chen et al., 2020). The model is then exposed to 15M image–caption pairs from YFCC15M (Thomee et al., 2016) in standard self-supervised fashion, allowing it to scale up while remaining anchored to perceptual structure.

Without any post-hoc tuning, this two-stage pipeline yields improvements across a variety of datasets and benchmarks including top 1, top 5, and retrieval. By transforming random seeds into perceptual seeds, we convert an often ignored source of variance into a principled inductive bias and set the trajectory of representation learning on a more human-aligned course from the very first gradient step.

## 2 PREVIOUS WORK

**Contrastive Vision–Language Pretraining.** Large-scale image–text contrastive learning frameworks have emerged as a foundation for vision–language models. CLIP (Radford et al., 2021) and ALIGN (Jia et al., 2021) demonstrated that pretraining visual encoders on web-scale image–caption data yields representations with strong zero-shot transfer performance. Subsequent works refined this paradigm; for example, Zhai et al. (2022) found that starting from a high-quality pretrained image encoder significantly improves training efficiency and final accuracy. Their LiT approach locked a pretrained ViT model and learned only a text tower, achieving a remarkable 85.2% zero-shot ImageNet accuracy—surpassing CLIP by over 8%—and highlighting the importance of initialization on downstream performance. However, these contrastive methods do not incorporate human perceptual knowledge during pretraining, instead relying on noisy web text as a proxy for semantics (He et al., 2020; Li et al., 2021; Bao et al., 2022). Our work departs by injecting an explicit human perceptual signal at the outset of pretraining.

**Post-hoc Behavioral Alignment.** Because standard models may not align with fine-grained human perception, a growing line of research augments pretrained representations with human behavioral

data *after* the main training phase. For instance, Muttenthaler *et al.* propose a global–local transform that linearly aligns a model's embedding space to human similarity judgments while preserving local structure, substantially improving few-shot and anomaly detection performance (Muttenthaler et al., 2023). In a similar vein, Zhao *et al.* fine-tune CLIP on 66-dimensional human behavioral embeddings (SPoSE descriptors) to produce CLIP-HBA, a model significantly more aligned with human judgments and even neural responses (Zhao et al., 2025). Sundaram *et al.* fine-tune vision backbones on human perceptual triplet judgments, yielding improved counting, segmentation, and instance-retrieval performance while largely preserving other benchmark scores (Sundaram et al., 2024). These studies confirm that introducing human perceptual structure can enhance model interpretability and task transfer—but they also note that naive alignment can distort a model's learned space, necessitating careful regularization or architectural constraints.

**Incorporating Human Perceptual Structure.** A few works have sought to bake human perceptual priors into the training process itself. Dong *et al.* introduced PeCo, a perceptual codebook that enforces that similar images map to nearby tokens during Vision Transformer pretraining, yielding more semantically meaningful tokens and +1.3% ImageNet accuracy over a BEiT baseline (Dong et al., 2022). Another line of research found that adversarially robust vision–language models (Robust CLIP) induce feature spaces that better mirror human perceptual judgments—even without any human labels—yielding more robust and interpretable perceptual metrics (Schlarmann et al., 2024; Croce et al., 2025b). Crucially, however, no prior work has directly integrated supervised human perceptual data into the core pretraining loop of vision–language models.

**Our Contribution: Perceptual-Initialization.** We build on these insights but depart by using human perceptual knowledge as a starting point for web-scale training. To our knowledge, ours is the first approach to utilize human triplet judgments to initialize a vision–language model's parameters prior to conventional image–text pretraining. This perceptual initialization infuses a human-aligned inductive bias from the very beginning, seeding the model's representation space before exposing it to 15 million image–text pairs (YFCC15M (Gu et al., 2024)). Our zero-shot evaluations across 23 of the 29 datasets confirm that this strategy yields systematically better generalization, opening a new paradigm for pretraining with human-based initialization.

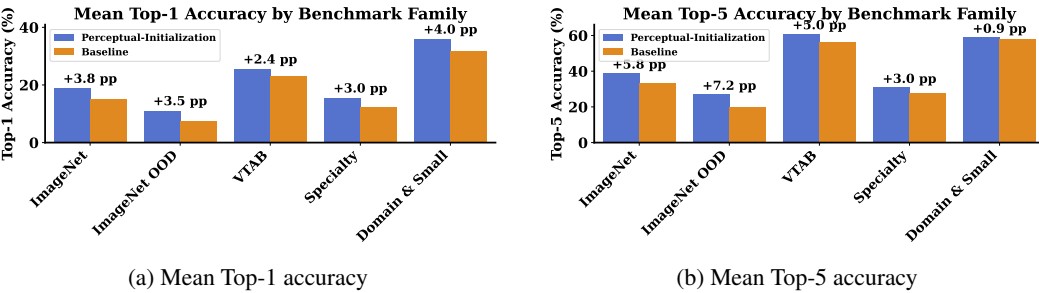

(a) Mean Top-1 accuracy        (b) Mean Top-5 accuracy

Figure 2: **Perceptual-Initialization yields consistent zero-shot gains across all benchmark families.** **(a)** Mean Top-1 accuracy and **(b)** mean Top-5 accuracy after 32 epochs of YFCC15M pre-training. PI surpasses the web-only baseline for every family—ImageNet, ImageNet-OOD, VTAB, Fine-grained & Specialty, and Domain & Small. Numbers above the bars denote the average lift in percentage points (pp). Overall, PI improves performance on 23 of 29 individual classification benchmarks.

## 3 THE PERCEPTUAL-INITIALIZED PRETRAINING PARADIGM

We propose a Perceptually-Initialized pretraining paradigm for vision-language models, specifically the Contrastive Language-Image Pre-training (CLIP) model (Radford et al., 2021) with a Vision Transformer (ViT-B/32) backbone (Dosovitskiy et al., 2021). Instead of applying human perceptual alignment as a post-hoc fine-tuning step, our approach integrates human perceptual judgments at the initial stage of representation learning. This paradigm consists of two sequential stages: first, initializing the vision encoder by training it on human similarity judgments, followed by a second stage of conventional large-scale contrastive pretraining on image-text pairs from the web.

## 3.1 STAGE 1: PERCEPTUAL INITIALIZATION OF THE VISION ENCODER

**Dataset.** We utilize the NIGHTS dataset, which comprises approximately 20,000 image triplets. Each triplet $(x, \tilde{x}_0, \tilde{x}_1)$ consists of a reference image $x$ and two synthetically generated variations, $\tilde{x}_0$ and $\tilde{x}_1$. These triplets are annotated with two-alternative forced-choice (2AFC) human similarity judgments, $y \in \{0, 1\}$, indicating which variation image humans perceived as more similar to the reference $x$. The dataset focuses on mid-level visual properties such as pose, layout, shape, and color, while maintaining roughly the same semantic content within a triplet (Fu et al., 2023).

**Model and Objective.** For this stage, we employ a CLIP model architecture using a ViT-B/32 for the vision encoder and a custom Transformer-based text encoder (Radford et al., 2021). Crucially, during this perceptual initialization stage, only the parameters $\theta_v$ of the vision encoder $f_{\theta_v}(\cdot)$ are trained.

We train the vision encoder using a triplet contrastive loss. Given a triplet, the vision encoder produces feature embeddings $f_{\theta_v}(x)$, $f_{\theta_v}(\tilde{x}_0)$, and $f_{\theta_v}(\tilde{x}_1)$. The dissimilarity (distance) between two images, say $(x, \tilde{x}_0)$, is measured by the cosine distance between their respective image features:

$$d(x, \tilde{x}_0) = 1 - \frac{f_{\theta_v}(x) \cdot f_{\theta_v}(\tilde{x}_0)}{\|f_{\theta_v}(x)\| \|f_{\theta_v}(\tilde{x}_0)\|}. \tag{1}$$

The alignment loss encourages the model to match human preferences, defined as (Sundaram et al., 2024):

$$\mathcal{L}_{\text{perceptual}}(\theta_v) = \mathbb{E}_{(x, \tilde{x}_0, \tilde{x}_1, y) \sim \mathcal{D}_{\text{NIGHTS}}} \left[ \max(0, m - \Delta d \cdot \bar{y}) \right], \tag{2}$$

where $\Delta d = d(x, \tilde{x}_0) - d(x, \tilde{x}_1)$, $\bar{y}$ maps the human judgment $y \in \{0, 1\}$ to $\{-1, 1\}$ (specifically, if $y = 0$ meaning $\tilde{x}_0$ is more similar, $\bar{y} = -1$; if $y = 1$ meaning $\tilde{x}_1$ is more similar, $\bar{y} = 1$). The margin $m$ is set to 0.05, following (Sundaram et al., 2024). This loss minimizes the distance between the reference and the human-preferred variation, while maximizing the distance to the less-preferred variation.

The vision encoder is trained for 32 epochs on the NIGHTS dataset using the AdamW optimizer.

## 3.2 STAGE 2: JOINT VISION-LANGUAGE PRETRAINING ON WEB-SCALE DATA

Following perceptual initialization, the full CLIP model undergoes standard contrastive pretraining on a large-scale web dataset.

**Dataset.** We use YFCC15M a subset of the YFCC100M dataset (Thomee et al., 2016) filtered by (Gu et al., 2024), consisting of approximately 15 million image-text pairs.

**Model and Objective.** The vision encoder, initialized with parameters $\theta_v$ from Stage 1, is unfrozen. Simultaneously, the text encoder—initialized with random parameters $\theta_t$—is also unfrozen. Both encoders are trained concurrently using the standard symmetric InfoNCE loss (van den Oord et al., 2018), as originally used for CLIP model training (Radford et al., 2021; He et al., 2020). The learnable logit scaling parameter, $\tau$, is also optimized during training.:

$$\mathcal{L}_{\text{CLIP}}(\theta_v, \theta_t, \tau) = -\frac{1}{2N} \sum_{i=1}^{N} \left( \log \frac{\exp(s(I_i, T_i)/\tau)}{\sum_{j=1}^{N} \exp(s(I_i, T_j)/\tau)} + \log \frac{\exp(s(T_i, I_i)/\tau)}{\sum_{j=1}^{N} \exp(s(T_i, I_j)/\tau)} \right), \tag{3}$$

where $I_i$ and $T_i$ are the image and text features for the $i$-th pair in a batch of size $N$, and $s(\cdot, \cdot)$ denotes cosine similarity.

The full CLIP model is trained for 32 epochs on the YFCC15M dataset using the AdamW optimizer (Loshchilov & Hutter, 2019).

## 3.3 COMPARATIVE MODELS

To evaluate the efficacy of our Human-First pretraining paradigm, we compare it against two key alternative approaches:

**Baseline YFCC15M Pretraining.**    This model serves as our primary baseline. A CLIP ViT-B/32 model, with both vision and text encoders randomly initialized, is trained from scratch on the YFCC15M dataset for 32 epochs using the InfoNCE loss (Equation 3) and the AdamW optimizer. This setup mirrors standard CLIP pretraining.

**Perceptual Fine-tuning.**    This approach aligns with prior work that applies perceptual alignment as a subsequent fine-tuning step documented by Sundaram et al. (2024). We utilized the baseline model described above. And then fine-tuned on the NIGHTS dataset for 8 epochs using the perceptual triplet loss (Equation 2) with an AdamW optimizer. Crucially, during this fine-tuning stage, only the Query, Key, and Value (QKV) projection matrices within each attention block of the ViT-B/32 vision encoder are unfrozen and updated. All other parameters of the vision encoder, the entire text encoder, and the logit scale remain frozen with their YFCC15M-trained weights.

## 3.4    IMPLEMENTATION DETAILS

Across all experiments, we use a CLIP ViT-B/32 architecture. The AdamW optimizer is used throughout with a learnable logit scale initialized to $\ln(100)$ for Stage 2 and baseline training. Images are processed at $224 \times 224$ resolution with consistent augmentations across all training scenarios, including random crops, color jitter, grayscale, Gaussian blur, and horizontal flips, followed by normalization using CLIP's standard values.

Stage 1 Initialization is extremely lightweight: a full 32-epoch run completes in roughly 30 min on the 6 × A100 node, amounting to $\sim 3$ GPU-hours in total. Stage 2 and the baseline YFCC15M pre-training share identical hyperparameters, same hardware (6 × A100), batch size (30,720), and duration (32 epochs). Each Stage 2 epoch takes $\sim 20$ wall-clock hours, i.e. $\sim 120$ GPU-hours per epoch, for a total of $\sim 3.8\,\mathrm{k}$ GPU-hours over the 32-epoch run.

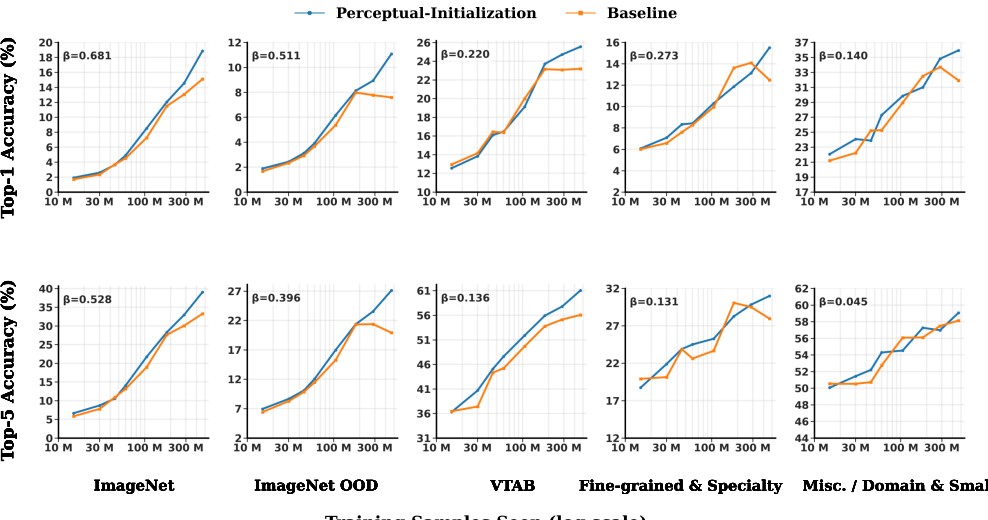

Figure 3: **Zero-shot classification scaling results.** Top-1 accuracy (top row) and Top-5 accuracy (bottom row) are shown for five benchmark families—ImageNet, ImageNet OOD, VTAB, Fine-grained & Specialty, and Misc./Domain & Small—plotted against the log-scale of training samples seen (10 M → 300 M) over total of 32 training epochs. The blue curve denotes our Perceptual-Initialization pipeline (NIGHTS20k → YFCC15M) and the orange curve the web-only baseline (YFCC15M). For each curve, we compute $\beta$ as the slope of a log–log linear fit between training size and performance. Across all families, Perceptual-Initialization attains higher initial accuracy and exhibits larger scaling exponents $\beta$, reflecting steeper performance gains as more data are ingested.

Table 1: **Zero-shot classification results by bucket.** Values show Top-1 and Top-5 accuracies for Perceptual-Initialization (PI@K), the web-only baseline (Base@K), and Perceptual Fine-Tuning (PFT@K). Bold indicates the best performance per metric. We include PFT's failure cases where human-aligned finetuning disrupts the model's image–text alignment and yields near random accuracy to illustrate the breakdown of this approach.

| Dataset | Task | #Test | #Cls | Ours@1 | Base@1 | △@1 | Ours@5 | Base@5 | △@5 |
|---|---|---|---|---|---|---|---|---|---|
| *ImageNet* | | | | | | | | | |
| ImageNet-1k 52 | Visual recog. | 50 000 | 1 000 | **18.9** | 15.1 | +3.8 | **39.0** | 33.3 | +5.7 |
| *ImageNet OOD* | | | | | | | | | |
| ImageNet-A 26 | Visual recog. | 7 500 | 200 | **5.3** | 4.0 | +1.3 | **19.0** | 16.0 | +3.0 |
| ImageNet-O 26 | Visual recog. | 2 000 | 200 | **21.6** | 14.3 | +7.3 | **46.2** | 31.9 | +14.3 |
| ImageNet-R 25 | Visual recog. | 30 000 | 200 | **15.0** | 10.3 | +4.7 | **34.7** | 24.6 | +10.1 |
| ImageNet-Sketch 62 | Visual recog. | 50 889 | 1 000 | **4.1** | 3.0 | +1.1 | **11.7** | 9.1 | +2.6 |
| ImageNet-V2 51 | Visual recog. | 10 000 | 1 000 | **13.1** | 8.3 | +4.8 | **30.6** | 20.0 | +10.6 |
| ObjectNet 2 | Visual recog. | 18 574 | 113 | **7.3** | 5.5 | +1.8 | **20.5** | 17.8 | +2.7 |
| *VTAB* | | | | | | | | | |
| CIFAR-100 32 | Visual recog. | 10 000 | 100 | **35.9** | 33.0 | +2.9 | **67.5** | 64.9 | +2.6 |
| Caltech-101 15 | Object recog. | 6 085 | 102 | 44.7 | **47.9** | -3.2 | **82.7** | 80.9 | +1.8 |
| CLEVR-Dist. 28 | Distance pred. | 15 000 | 6 | 15.8 | **16.1** | -0.3 | 90.7 | **91.0** | -0.3 |
| CLEVR-Count 28 | Counting | 15 000 | 8 | **16.1** | 11.5 | +4.6 | 61.8 | **65.3** | -3.5 |
| KITTI-CVD 19 | Distance pred. | 711 | 4 | **32.1** | 31.5 | +0.6 | — | — | — |
| DTD 8 | Texture cls. | 1 880 | 47 | **14.3** | 10.4 | +3.9 | **34.4** | 28.0 | +6.4 |
| EuroSAT 24 | Satellite recog. | 5 400 | 10 | **24.7** | 19.6 | +5.1 | **76.2** | 69.0 | +7.2 |
| Flowers-102 46 | Flower recog. | 6 149 | 102 | **26.7** | 18.7 | +8.0 | **52.3** | 40.9 | +11.4 |
| Oxford-IIIT Pet 48 | Pet cls. | 3 669 | 37 | **17.2** | 11.6 | +5.6 | **38.8** | 29.1 | +9.7 |
| RESISC45 6 | Remote-sens. | 6 300 | 45 | **17.6** | 15.9 | +1.7 | **45.2** | 37.1 | +8.1 |
| SVHN 45 | Digit recog. | 26 032 | 10 | 11.0 | **11.3** | -0.3 | **61.0** | 54.5 | +6.5 |
| PCAM 61 | Histopath. cls. | 32 768 | 2 | 50.5 | **50.8** | -0.3 | — | — | — |
| *Fine-grained & Specialty* | | | | | | | | | |
| Stanford Cars 31 | Vehicle recog. | 8 041 | 196 | **1.7** | 1.5 | +0.2 | **6.9** | **6.9** | +0.0 |
| Food-101 3 | Food recog. | 25 250 | 101 | **12.1** | 8.3 | +3.8 | **35.8** | 26.0 | +9.8 |
| FGVC-Aircraft 41 | Aircraft recog. | 3 333 | 100 | 1.6 | **1.7** | -0.1 | **6.5** | 5.7 | +0.8 |
| PASCAL VOC 07 14 | Object recog. | 14 976 | 20 | **46.6** | 38.4 | +8.2 | **74.8** | 73.3 | +1.5 |
| *Misc. / Domain & Small* | | | | | | | | | |
| CIFAR-10 32 | Visual recog. | 10 000 | 10 | **69.5** | 62.4 | +7.1 | **95.9** | 95.7 | +0.2 |
| Country211 63 | Geolocation | 21 100 | 211 | **3.5** | 3.0 | +0.5 | **11.3** | 10.4 | +0.9 |
| GTSRB 57 | Traffic-sign recog. | 12 630 | 43 | **7.0** | 5.9 | +1.1 | **35.0** | 32.8 | +2.2 |
| MNIST 33 | Digit recog. | 10 000 | 10 | **12.4** | 11.6 | +0.8 | **55.1** | 55.0 | +0.1 |
| Rendered-SST2 56 | Sentiment cls. | 1 821 | 2 | **49.9** | **49.9** | +0.0 | — | — | — |
| STL-10 9 | Visual recog. | 8 000 | 10 | **73.2** | 58.6 | +14.6 | **98.0** | 96.8 | +1.2 |

## 4 RESULTS

### 4.1 ZERO-SHOT CLASSIFICATION

**Benchmarks and Setup.** We assess zero-shot classification performance on a comprehensive suite of 29 datasets. To facilitate a nuanced analysis across various visual domains and task complexities, these datasets are categorized into five distinct families: ImageNet, ImageNet Out-of-Distribution (OOD), VTAB, Fine-grained & Specialty, and Miscellaneous / Domain & Small. The specific datasets (and evaluations) constituting each family are enumerated in Table 1. This grouping strategy is adopted to provide a structured understanding of model generalization across different data distributions, akin to methodologies used in large-scale evaluations like DataComp (Gadre et al., 2023). We report Top-1 and Top-5 accuracy for all classification tasks.

**Scaling Laws by Benchmark Family.** Figure 3 disaggregates the scaling behaviour of our Perceptual-Initialization model (blue) versus the web-only baseline (orange) across five benchmark families. The top row reports Top-1 accuracy, and the bottom row reports Top-5 accuracy, each plotted against the log-scale count of YFCC15M training samples. Across all families, Perceptual-Initialization either outperforms or matches the baseline at every scale. The power-law exponents ($\beta$) calcualted as the slope of a log–log linear fit between training size and performance, shown in each panel measure the steepness of these gains and are uniformly higher for Perceptual-Initialization—indicating faster improvement as more data are ingested. Crucially, the method establishes a sizeable head start on ImageNet and ImageNet-OOD and maintains (or widens) that

margin throughout pre-training, highlighting the broad utility of embedding a perceptual prior from the outset. Extended per dataset scaling results are listed in the App. C (Supplementary).

**Aggregated Performance Gains.** The cumulative advantages of perceptual initialization are concisely summarized in Figure 2, which presents the mean Top-1 (Fig. 2a) and Top-5 (Fig. 2b) accuracy improvements, averaged across the datasets within each respective family, upon completion of 32 training epochs. For Top-1 accuracy, our model demonstrates notable gains over baseline across all five benchmark families: +3.8 percentage points (pp) on ImageNet, +3.5 pp on ImageNet OOD, +2.4 pp on VTAB, +3.0 pp on Fine-grained & Specialty, and +4.0 pp on Misc./Domain & Small. Consistent positive outcomes are also evident for Top-5 accuracy, with improvements of +5.8 pp (ImageNet), +7.2 pp (ImageNet OOD), +5.0 pp (VTAB), +3.0 pp (Fine-grained & Specialty), and +0.9 pp (Misc./Domain & Small). These results compellingly indicate that seeding models with human perceptual priors fosters systematically enhanced generalization capabilities on a wide spectrum of unseen classification tasks. As noted in the caption of Figure 2, our approach surpasses the baseline on 23 out of 29 Top-1 classification benchmarks (with 1 tie and 5 losses), highlighting the widespread nature of the improvements.

## 4.2 ZERO-SHOT RETRIEVAL

Table 2: Retrieval performance (Recall@K)

| | Flickr 1K Test | | | | MS-COCO 2014 5K Test | | | |
| | Image→Text | | Text→Image | | Image→Text | | Text→Image | |
| Model | R@1 | R@5 | R@1 | R@5 | R@1 | R@5 | R@1 | R@5 |
| --- | --- | --- | --- | --- | --- | --- | --- | --- |
| Baseline (YFCC) | 14.2 | 32.9 | 24.3 | 51.0 | 7.3 | 19.7 | 14.7 | 33.1 |
| Human-first (ours) | **21.3** | **45.3** | **31.6** | **60.7** | **10.0** | **25.2** | **18.0** | **38.8** |

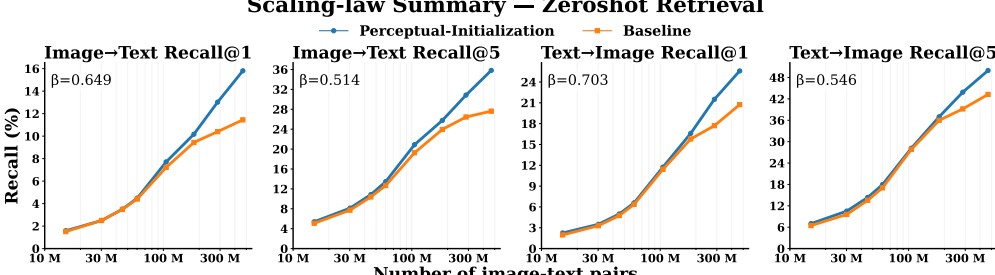

Figure 4: **Retrieval Tasks Scaling Results.** Recall@1 and Recall@5 are plotted (log-scale, number of image–text pairs seen) over successive epochs on YFCC15M for two retrieval directions: (a) Image → Text R@1, (b) Image → Text R@5, (c) Text → Image R@1, and (d) Text → Image R@5. The blue curves show our proposed perceptual initialization method, while the orange curves represent the conventional web-scale baseline. A performance gap between the two methods becomes apparent after just a few epochs and grows steadily as more data is ingested, underscoring the strong and increasing advantage of our approach with larger training-sample scales.

**Benchmarks and setup.** Zero-shot image–text retrieval is assessed on two standard benchmarks—MS-COCO Captions (Lin et al., 2015) and Flickr30k (Young et al., 2014). For each benchmark we report both retrieval directions, image → text (I→T) and text → image (T→I), using Recall@1 (R@1) and Recall@5 (R@5).

**Scaling-law Analysis.** Figure 4 plots four curves: I→T R@1, I→T R@5, T→I R@1, and T→I R@5, each as a function of the log-scale number of image–text pairs seen during YFCC15M pre-training. Across all metrics and scales the Perceptual Initialization model (blue) consistently surpasses or matches the web-only baseline (orange). The power-law exponents ($\beta$), annotated on each subplot are uniformly higher for our method, signaling steeper performance gains as more data are ingested.

Together with the classification results, these curves show that injecting a perceptual prior not only yields an early lead but also preserves a stronger scaling trend throughout training.

**Comparison to human-later fine-tuning.** For completeness, we replicated the post-hoc perceptual fine-tuning protocol from Sundaram et al. (2024), running eight additional epochs of NIGHTS triplet supervision after the 32-epoch YFCC15M pre-training under identical Stage-2 hyperparameters. Although this increased NIGHTS validation accuracy to 91%, it catastrophically disrupted the learned image–text alignment: zero-shot classification family means fell sharply, and retrieval collapsed (e.g., COCO I↦T R@1 14.2% → 1.3%). Full per-dataset tables are provided in App. D (Supplementary). These results indicate that perceptual supervision is most effective when applied at initialization, rather than as a late-stage retrofit.

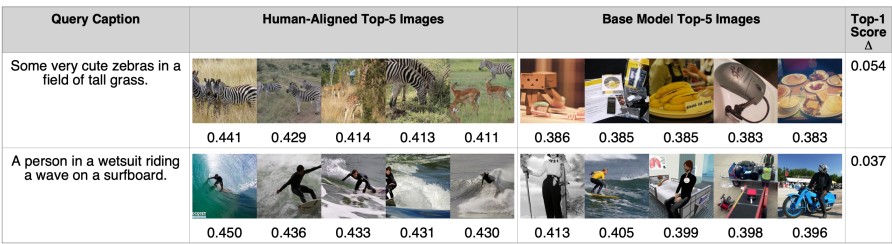

Qualitative Comparison: Human-Aligned vs. Base Model (Top-5)

| Image | Ground Truth | Human-Aligned Top-5 (Score) | Base Model Top-5 (Score) | Top-1 Score Δ |
|---|---|---|---|---|
|  | A black and white picture of a stop sign. **A black-and-white photo of a stop sign by some grass.** A stop sign stands on a pathway near a wooded area. **Black and white photo of a stop sign on a rural street.** A stop sign that is in the middle of nowhere. | **Black and white photo of a stop sign on a rural street. (0.454)** The cars has stopped at the red stop sign (0.417) **A black-and-white photo of a stop sign by some grass. (0.408)** This is a stop sign at the end of a road in front of a fence. (0.405) A stop sign at the end of a dead end road (0.403) | Roadside traffic sign that posting the speed limit and the direction of upcoming curve direction. (0.417) A 20mph speed limit sign at a tree lined intersection (0.417) A speed limit sign on the side of a neighborhood street (0.416) A street sign above a speed limit sign on a rural street. (0.412) A traffic sign near a high grass field near a road. (0.411) | 0.037 |
|  | A young man riding a skateboard up a black ramp. A skateboarding boy is about to go onto the red skate bar. A skateboarder starting a jump on a homemade ramp. A person standing on a skateboard and performing a stunt on a platform in the street. **A man is riding a skateboard up a ramp on a street in front of a truck.** | A worker driving a cart pulling a trailer loaded with cargo. (0.411) A man holding glass near a pick up truck on the street. (0.411) **A man is riding a skateboard up a ramp on a street in front of a truck. (0.407)** Two boys moving along outside during the day. one of them has a skateboard. (0.404) A man is trying to pull off a skateboarding trick on his ramp. (0.398) | A couple of men are loading a truck with glass (0.407) Two women eat chili dogs on a city sidewalk. (0.404) A woman riding a bike down the street (0.403) Two boys getting ready to go down the skateboard ramp on their skateboards. (0.402) A man and woman loading a surfboard on a motorcycle outside with other riders nearby (0.402) | 0.004 |

(a) Image-to-Text Retrieval

| Query Caption | Human-Aligned Top-5 Images | Base Model Top-5 Images | Top-1 Score Δ |
|---|---|---|---|
| Some very cute zebras in a field of tall grass. | 0.441   0.429   0.414   0.413   0.411 | 0.386   0.385   0.385   0.383   0.383 | 0.054 |
| A person in a wetsuit riding a wave on a surfboard. | 0.450   0.436   0.433   0.431   0.430 | 0.413   0.405   0.399   0.398   0.396 | 0.037 |

(b) Text-to-Image Retrieval

Figure 5: **Qualitative comparison of zero-shot retrieval.** *(a) Image→Text:* For two query images, we list the ground-truth captions (left) and the top-5 captions returned by each model, together with their cosine similarity scores (higher is better). Ground-truth matches are highlighted in **bold**. The PI model retrieves the correct caption in every case, with higher cosine similarity scores and larger Top-1 margins (Δ) compared to the baseline. *(b) Text→Image:* For two query captions, we show the top-5 retrieved images per model, with similarity scores beneath each thumbnail. In the first example, only the PI model retrieves zebras in the top ranks and secures a significantly higher Top-1 score (0.441 vs. 0.386). In the second example, both models retrieve surfing scenes, yet the PI model still secures a better Top-1 score.

**Qualitative examples.** Figure 5 visualizes how our model behaves compared with a from-scratch baseline on two representative zero-shot retrieval tasks image→text and text→image. Across both directions, the PI model consistently ranks the ground-truth item higher and with a larger similarity margin, indicating that the human-derived triplet supervision indeed steers the representation toward

more semantically faithful matches. Extended qualitative examples, including failure cases are listed in the App. E (Supplementary).

### 4.3 GENERALITY ACROSS ARCHITECTURES

To test whether PI is tied to a specific backbone, we ran two exploratory trainings beyond ViT-B/32. A CLIP ResNet-50 encoder perceptually initialized on NIGHTS and trained for 32 epochs on YFCC15M replicates the trend, improving mean Top-1 accuracy by +4.09 pp against a randomly-initialized ResNet-50 baseline. A large-capacity ViT-L/14 model shows PI gains after only 16 epochs, with early checkpoints already out-performing the size-matched baseline on 23/29 classification benchmarks. These preliminary results suggest that PI provides a backbone agnostic inductive bias that benefits both convolutional and transformer families without additional tuning.

## 5 DISCUSSION

Embedding human perceptual structure at the very start of pre-training produces quantitative and qualitative benefits that conventional self-supervised pipelines do not attain. With perceptual priors as an initialization, zero-shot accuracy surpasses the from-scratch baseline on 23 of 29 evaluation datasets (79%) 1, overall across all families of evaluations, and it does so early enough to translate into meaningful compute savings. These findings reinforce prior evidence that deliberate weight initialization can steer convergence more decisively than many downstream hyper-parameters(Mehrer et al., 2020; Picard, 2021; Bouthillier et al., 2021). Crucially, we integrate the perceptual prior jointly with contrastive learning rather than performing fine-tuning as a costly second stage, thereby preserving the simplicity of a single-stage workflow.

A natural question is not only how much but also which behavioral data suffice. We will train identical models on 1%–100% of the Nights triplets, charting zero-shot accuracy to locate the fraction at which gains become statistically reliable. The same sweep will be run with alternative priors—object-level THINGS/SPoSE and low/mid-level BAPPS dataset(Zhang et al., 2018; Hebart et al., 2019; 2020)—under a fixed stage-1 compute budget. Comparing these data-efficiency curves will reveal both the minimal data budget and the most informative behavioral source, albeit at non-trivial computational cost because each model must be evaluated across training.

Beyond purely behavioral priors, recent work shows that fine-tuning CLIP with neural embeddings can personalize representations to individual brains (Zhao et al., 2025). Our results open the door to joint behavioral–neural pretraining in which MEG- or fMRI-derived embeddings act as an additional perceptual channel, enriching the representation space while keeping compute manageable thanks to earlier convergence (Cichy et al., 2016; Schrimpf et al., 2018; Kaniuth & Hebart, 2022; Oota et al., 2024).

Extending the idea across modalities promises still larger dividends. Audio similarity judgments or cross-modal correspondence tasks could supply complementary priors that interact supra-additively with vision, much as synergistic gains have been reported for robust CLIP adversarial fine-tuning (Schlarmann et al., 2024).

Several challenges nevertheless remain. Stimulus representativeness is the first with even large triplet sets oversampling frequent objects and viewpoints, leaving rare or long-tail concepts sparsely covered. Building behaviorally balanced libraries via targeted synthesis, active sampling, or human-in-the-loop curation can reduce blind spots. High-quality behavioral embeddings beyond vision are still scarce with large-scale, carefully designed datasets for audition, haptic, or olfaction virtually non-existent (Liu et al., 2022; Li et al., 2024b;a). Collecting or transferring such priors is essential for truly multimodal PI. Behavioral bias, finally, persists even in well-sampled datasets. Human judgments reflect demographic, cultural, and contextual biases that can propagate into the model's decision boundary. Balanced sampling across populations, adversarial debiasing objectives, and fairness-aware loss terms therefore remain critical directions for future work.

Taken together, these findings provide the first large-scale evidence that beginning with you, placing human perception at the origin of representation learning, produces models that are faster, better aligned, and more versatile. We hope this work catalyzes broader exploration of perceptual priors across architectures, modalities, and levels of biological fidelity.

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
