# SUPPLEMENTARY MATERIAL FOR
# BEGINNING WITH YOU: PERCEPTUAL-INITIALIZATION IMPROVES VISION-LANGUAGE REPRESENTATION AND ALIGNMENT

## A    IMPLEMENTATION DETAILS

This appendix provides further details on our model architecture and the training protocols employed for Perceptual Initialization (PI) and related experiments, supplementing the information presented in the main paper. Full training code for PI is available in our public repository[1].

### A.1    ARCHITECTURAL FOUNDATION AND BASELINE CLIP PRINCIPLES

Table 1: Training hyper-parameters and compute budget for each stage.

| Scale | Model | Train MACs | GPUs | #Samples | LR | $\beta_2$ | Warm-up | Batch |
|-------|-------|-----------|------|----------|-----|-----------|---------|-------|
| PI Stage 1 | ViT-B/32 | $1.15 \times 10^{16}$ | 6 | $4.35 \times 10^5$ | $5 \times 10^{-4}$ | 0.999 | 150 | 768 |
| PI Stage 2 | ViT-B/32 | $7.10 \times 10^{18}$ | 6 | $4.80 \times 10^8$ | $5 \times 10^{-4}$ | 0.98 | 2 500 | 30 720 |
| Baseline | ViT-B/32 | $7.10 \times 10^{18}$ | 6 | $4.80 \times 10^8$ | $5 \times 10^{-4}$ | 0.98 | 2 500 | 30 720 |
| PFT | ViT-B/32 (QKV) | $2.87 \times 10^{15}$ | 6 | $1.09 \times 10^5$ | $3 \times 10^{-4}$ | 0.999 | 150 | 96 |

Our work builds upon the core concepts of Contrastive Language-Image Pre-training (CLIP) (Radford et al., 2021), which utilizes a dual-encoder architecture to learn joint representations of images and text. The original CLIP framework typically involves an image encoder (either a ResNet variant or a Vision Transformer) and a Transformer text encoder (He et al., 2015; Dosovitskiy et al., 2021; Vaswani et al., 2023). These encoders project images and their corresponding textual descriptions into a shared embedding space. Alignment is achieved by maximizing the similarity of true image-text pairs while minimizing it for incorrect pairings, often using a symmetric InfoNCE loss (Gutmann & Hyvärinen, 2010; van den Oord et al., 2018). This pre-training is performed on vast web-scale datasets of image-text pairs (Thomee et al., 2016; Schuhmann et al., 2021; Desai et al., 2021; Radford et al., 2021; Schuhmann et al., 2022).

For our experiments, including the Perceptual Initialization stages, we employ a ViT based architecture for the image encoder and a Transformer-based text encoder. Specifically, our ViT model is a ViT-B/32, characterized by 12 layers, a hidden width of 768, and 12 attention heads with each head width of 64. It processes $224 \times 224$ images divided into $32 \times 32$ patches. The architecture includes patch embeddings, a learnable class [CLS] token, learnable positional embeddings, and standard Transformer blocks consisting of multi-head self-attention and MLP sub-layers. Layer normalization is applied, and GELU activation functions are used (Hendrycks & Gimpel, 2023). The text encoder is a 12-layer Transformer with a hidden width of 512 and 8 attention heads, processing tokenized text sequences up to a context length of 77 tokens, using a vocabulary of 49,408 BPE tokens. Both encoders output 512-dimensional embeddings, which are $L_2$-normalized before similarity computation. A learnable temperature parameter scales the logits in the contrastive loss. Our implementation is adapted from Cherti et al. (2023) and Li et al. (2022); please refer to our public repository for further details.

---

[1]https://anonymous.4open.science/r/perceptual-pretrain-5D4F/

## A.2 Perceptual Initialization (PI) Training Strategy

**PI Stage 1: Perceptual Initialization on NIGHTS.** The first stage of PI aims to imbue the vision encoder with foundational human perceptual understanding. We train the ViT-B/32 image encoder using the NIGHTS dataset (see Appendix B.1 for details). This dataset, comprising image triplets, facilitates learning through a triplet similarity objective. The model is trained to minimize a triplet loss with a specified margin (0.05 in our configuration, using cosine distance), encouraging it to learn visual features that align with human perceptual judgments of similarity. For this stage, we use an AdamW optimizer with a learning rate of $5 \times 10^{-4}$, $\beta_1 = 0.9$, and $\beta_2 = 0.999$. A cosine learning rate decay schedule is applied following a warm-up phase of 150 iterations. Training is conducted with a global batch size of 768 across 6 GPUs, processing approximately $4.35 \times 10^5$ image samples. Further details on computational budget (MACs) are provided in Table 1.

**PI Stage 2: Large-Scale Vision-Language Pre-training on YFCC15M.** Following perceptual initialization on NIGHTS, the initialized ViT-B/32 vision encoder, along with the text encoder, undergoes large-scale contrastive pre-training on the YFCC15M dataset (see Appendix B.2). This stage aligns the perceptually-grounded visual features with language representations using a standard InfoNCE contrastive loss. The optimizer remains AdamW, with a learning rate of $5 \times 10^{-4}$, $\beta_1 = 0.9$, but $\beta_2$ is set to 0.98 for this stage, aligning with practices for large-scale ViT training. The learning rate schedule includes a longer warm-up of 2,500 iterations followed by cosine decay. This stage uses a significantly larger global batch size of 30,720, distributed across 6 GPUs, and processes around $4.80 \times 10^8$ image-text samples. Computational details are summarized in Table 1.

## A.3 Comparative Baselines and Fine-tuning Implementations

To evaluate the efficacy of PI, we implement and train several comparative models.

**Baseline Training.** A baseline ViT-B/32 model is trained from scratch directly on the YFCC15M dataset using the same contrastive vision-language pre-training setup as PI Stage 2. This includes the same optimizer (AdamW, LR $5 \times 10^{-4}$, $\beta_2 = 0.98$), learning rate schedule (2,500 warm-up iterations, cosine decay), global batch size (30,720), and number of samples ($4.80 \times 10^8$) as detailed in Table 1. This allows for a direct comparison against a standard CLIP-style training approach without perceptual initialization.

**Perceptual Fine-Tuning (PFT).** We also investigate a Perceptual Fine-Tuning (PFT) approach following Sundaram et al. (2024). In this setup, a ViT-B/32 model, pre-trained on YFCC15M (analogous to our "Baseline" or the result of PI Stage 2), has its Query, Key, and Value (QKV) projection matrices within the self-attention mechanisms of its Transformer blocks fine-tuned on the NIGHTS dataset using the triplet loss objective. For PFT, we use an Adam optimizer with a learning rate of $3 \times 10^{-4}$, $\beta_1 = 0.9$, and $\beta_2 = 0.999$. The learning rate schedule incorporates a 150-iteration warm-up followed by cosine decay. Training is performed with a global batch size of 96 on 6 GPUs, processing approximately $1.09 \times 10^5$ samples from the NIGHTS dataset. Specifics are listed in Table 1.

## A.4 General Training Environment and Parameters

Across all training stages detailed above and in Table 1, we utilize BF16 mixed-precision training to accelerate computation and reduce memory footprint without significant loss in model performance. Gradient checkpointing is employed within the Transformer blocks during training to further manage memory consumption, allowing for larger models or batch sizes. Our implementations are built using PyTorch and leverage PyTorch Lightning for organizing the training loops, distributed training, and logging (Paszke et al., 2019; Falcon & the PyTorch Lightning Team, 2019). All models are trained with a weight decay, the specific value of which can vary by stage (e.g., 0.1 for YFCC15M stages, potentially different for NIGHTS stages as per detailed configurations). The number of training epochs for the primary stages (PI Stage 1, PI Stage 2, Baseline) is typically 32, though effective training duration is also a function of dataset size and batch size. Gradient norms are clipped to prevent exploding gradients, with specific clipping values adjusted per stage, 6.0 for YFCC15M pre-training, 3.0 for NIGHTS-based training/fine-tuning.

## B DATASETS REVIEW

### B.1 THE NIGHTS DATASET

The NIGHTS dataset comprises approximately 20,000 image triplets. Each triplet consists of a reference image and two synthetically generated variations. Human participants performed two-alternative forced-choice (2AFC) similarity judgments, indicating which variation was perceived as more similar to the reference image. These judgments yield a binary label for each triplet (Fu et al., 2023).

A defining feature of NIGHTS is its focus on mid-level visual properties such as color, pose, layout, and shape, while generally preserving the semantic content within each triplet. This design allows the dataset to capture fine-grained perceptual nuances often missed by metrics focused on pixel-level or high-level categorical similarity (Sundaram et al., 2024). The primary aim of NIGHTS is to enable the learning of novel dimensions of human visual similarity through synthetic data, specifically targeting "cognitively impenetrable" judgments—those that are rapid, consistent across individuals, and robust to changes in mental representation (Fu et al., 2023).

NIGHTS was generated using advanced text-to-image models, notably Stable Diffusion v1.4 (Rombach et al., 2022), to create synthetic image pairs with systematic perturbations along various visual dimensions. The generation process involved iterative filtering and prompting diffusion models with categories from established datasets like ImageNet (Russakovsky et al., 2015), CIFAR-10/100 (Krizhevsky & Hinton, 2009), Oxford 102 Flowers (Nilsback & Zisserman, 2008b), and Food-101 (Bossard et al., 2014a). For each triplet, multiple (up to 10) similarity judgments were collected, and the dataset was subsequently filtered to retain only unanimous examples, thereby enhancing sample quality by removing ambiguous cases . The reliability of these 2AFC judgments was further supported by Just Noticeable Difference (JND) studies. This synthetic generation approach offers scalability and precise control over visual attributes, which is crucial for studying human perception effectively (Fu et al., 2023).

### B.2 THE YFCC15M DATASET: LARGE-SCALE IMAGE-TEXT PRETRAINING

After perceptual initialization with NIGHTS, the YFCC15M dataset serves as the large-scale web dataset for the second stage of joint vision-language pretraining. YFCC15M is a curated subset of the Yahoo Flickr Creative Commons 100 Million (YFCC100M) dataset (Thomee et al., 2016), containing approximately 15 million image-text pairs. The original YFCC100M dataset is a vast multimedia collection of nearly 100 million photos and videos uploaded to Flickr between 2004 and 2014, each with associated metadata (e.g., title, description, tags) (Thomee et al., 2016).

The YFCC15M subset used in our work was filtered by Li et al. (2022) and enhanced by Gu et al. (2024). This enhancement involved a "diverse description generation framework" that leverages Large Language Models (LLMs) to combine and refine information from raw web image-text pairs, synthetically generated captions via OFA, and fine-grained detection tags from RAM++ (Wang et al., 2022; Huang et al., 2023; Gu et al., 2024). The goal of this framework is to mitigate noise inherent in web-crawled data and improve the semantic accuracy and richness of the image descriptions (Gu et al., 2024). This meticulous curation is vital for robust vision-language representation learning, especially for our second stage large-scale pretraining experiment.

### B.3 OTHER PERCEPTUAL DATASETS

While NIGHTS and YFCC15M are central to our work, the landscape of perceptual data is broader. Understanding other datasets helps contextualize our choices and highlights avenues for future exploration.

- **THINGS Dataset:** Contains 4.7 million pairwise similarity judgments for 1,854 everyday objects, along with interpretable SPoSE (Sparse Positive Similarity Embedding) embeddings (Hebart et al., 2019; 2023). It primarily uses an "odd-one-out" task to capture human judgments about object similarity and aims for a broad, systematic sampling of object representations. The SPoSE model derives interpretable behavioral dimensions (e.g., 66

dimensions from the full dataset) representing perceptual and conceptual object properties (Hebart et al., 2020).

- **BAPPS Dataset:** The Berkeley Adobe Perceptual Patch Similarity (BAPPS) dataset is a benchmark for evaluating perceptual image similarity metrics (Zhang et al., 2018). It uses a 2AFC test where participants identify which of two distorted images is more similar to a reference, focusing on low-level distortions. The Learned Perceptual Image Patch Similarity (LPIPS) metric was trained using BAPPS data (Zhang et al., 2018). M-BAPPS extends this with text descriptions for multimodal Image Quality Assessment (IQA) (You et al., 2024).

- **STUFF Dataset:** Focuses on material representations, comprising 1.87 million triplet similarity judgments on an image collection of 200 systematically sampled material categories (600 photos) (Schmidt et al., 2025). It aims to uncover latent dimensions of human material perception, with images depicting materials in their typical aggregate state.

The selection of NIGHTS for our Perceptual Initialization stage was intuitive. Its synthetic generation allows controlled variation of mid-level visual properties, and the 2AFC task on "cognitively impenetrable" judgments provides a clean, consistent human signal crucial for effective initialization (Fu et al., 2023; Sundaram et al., 2024). This aligns with our core hypothesis that "beginning with you" embedding human perceptual structure early yields more robust and aligned vision-language representations.

# C    COMPLETE SCALING CURVES

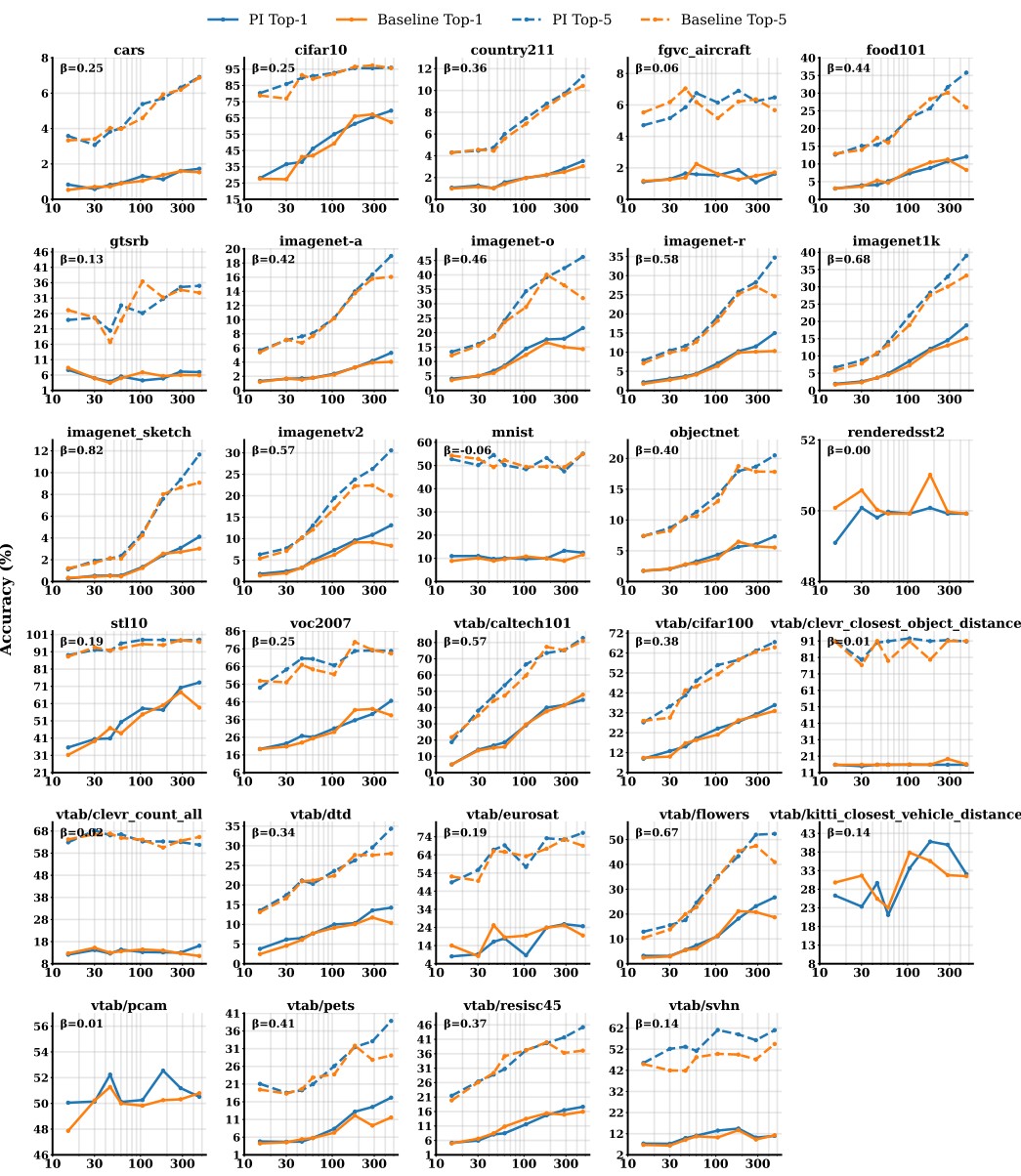

Figure 1: **Per-dataset scaling laws for zero-shot classification.** Top-1 and Top-5 accuracy for each of the 29 benchmark datasets are plotted against the log number of pre-training samples. Blue: Perceptual Initialization (PI). Orange: web-only baseline. The fitted power-law exponent $\beta$ annotates each curve. PI typically starts higher and scales faster, showing the benefit of injecting human perceptual priors at the very start of training.

# D  COMPLETE RESULTS TABLES

Table 2: **Zero-shot classification results by bucket.** Values show Top-1 and Top-5 accuracies for Perceptual-Initialization (PI@K), the web-only baseline (Base@K), and Perceptual Fine-Tuning (PFT@K). Bold indicates the best performance per metric. We include PFT's failure cases where human-aligned finetuning disrupts the model's image–text alignment and yields near random accuracy to illustrate the breakdown of this approach.

| Dataset | #Test | #Cls | PI@1 | Base@1 | PFT@1 | PI@5 | Base@5 | PFT@5 |
|---|---|---|---|---|---|---|---|---|
| *ImageNet* | | | | | | | | |
| ImageNet-1k 41 | 50 000 | 1 000 | **18.9** | 15.1 | 0.1 | **39.0** | 33.3 | 0.5 |
| *ImageNet OOD* | | | | | | | | |
| ImageNet-A 24 | 7 500 | 200 | **5.3** | 4.0 | 0.4 | **19.0** | 16.0 | 2.7 |
| ImageNet-O 24 | 2 000 | 200 | **21.6** | 14.3 | 0.4 | **46.2** | 31.9 | 1.6 |
| ImageNet-R 23 | 30 000 | 200 | **15.0** | 10.3 | 0.5 | **34.7** | 24.6 | 2.1 |
| ImageNet-Sketch 52 | 50 889 | 1 000 | **4.1** | 3.0 | 0.0 | **11.7** | 9.1 | 0.4 |
| ImageNet-V2 39 | 10 000 | 1 000 | **13.1** | 8.3 | 0.1 | **30.6** | 20.0 | 0.5 |
| ObjectNet 8 | 18 574 | 113 | **7.3** | 5.5 | 1.1 | **20.5** | 17.8 | 4.1 |
| *VTAB* | | | | | | | | |
| CIFAR-100 28 | 10 000 | 100 | **35.9** | 33.0 | 1.1 | **67.5** | 64.9 | 4.8 |
| Caltech-101 12 | 6 085 | 102 | 44.7 | **47.9** | 3.6 | **82.7** | 80.9 | 8.5 |
| CLEVR-Dist. 26 | 15 000 | 6 | 15.8 | **16.1** | 20.6 | 90.7 | **91.0** | 79.3 |
| CLEVR-Count 26 | 15 000 | 8 | **16.1** | 11.5 | 12.4 | 61.8 | **65.3** | 61.7 |
| KITTI-CVD 14 | 711 | 4 | **32.1** | 31.5 | 22.1 | — | — | — |
| DTD 5 | 1 880 | 47 | **14.3** | 10.4 | 1.4 | **34.4** | 28.0 | 11.8 |
| EuroSAT 21 | 5 400 | 10 | **24.7** | 19.6 | 13.7 | **76.2** | 69.0 | 58.4 |
| Flowers-102 34 | 6 149 | 102 | **26.7** | 18.7 | 1.5 | **52.3** | 40.9 | 3.7 |
| Oxford-IIIT Pet 36 | 3 669 | 37 | **17.2** | 11.6 | 2.4 | **38.8** | 29.1 | 13.5 |
| RESISC45 3 | 6 300 | 45 | **17.6** | 15.9 | 2.4 | **45.2** | 37.1 | 11.5 |
| SVHN 33 | 26 032 | 10 | 11.0 | **11.3** | 7.4 | **61.0** | 54.5 | 49.1 |
| PCAM 51 | 32 768 | 2 | 50.5 | **50.8** | 50.6 | — | — | — |
| *Fine-grained & Specialty* | | | | | | | | |
| Stanford Cars 27 | 8 041 | 196 | **1.7** | 1.5 | 0.7 | **6.9** | 6.9 | 2.9 |
| Food-101 2 | 25 250 | 101 | **12.1** | 8.3 | 0.9 | **35.8** | 26.0 | 4.5 |
| FGVC-Aircraft 32 | 3 333 | 100 | 1.6 | **1.7** | 1.3 | **6.5** | 5.7 | 4.8 |
| PASCAL VOC 07 10 | 14 976 | 20 | **46.6** | 38.4 | 3.9 | **74.8** | 73.3 | 29.1 |
| *Misc. / Domain & Small* | | | | | | | | |
| CIFAR-10 28 | 10 000 | 10 | **69.5** | 62.4 | 10.8 | **95.9** | 95.7 | 52.5 |
| Country211 54 | 21 100 | 211 | **3.5** | 3.0 | 0.5 | **11.3** | 10.4 | 2.2 |
| GTSRB 46 | 12 630 | 43 | **7.0** | 5.9 | 4.2 | **35.0** | 32.8 | 18.0 |
| MNIST 29 | 10 000 | 10 | **12.4** | 11.6 | 10.1 | **55.1** | 55.0 | 57.2 |
| Rendered-SST2 45 | 1 821 | 2 | 49.9 | 49.9 | **50.1** | — | — | — |
| STL-10 6 | 8 000 | 10 | **73.2** | 58.6 | 7.1 | **98.0** | 96.8 | 45.4 |

# E    EXTENDED QUALITATIVE EXAMPLES

## E.1    CLASSIFICATION TASKS

To better understand the differences between our proposed Perceptual Initialization, and the standard baseline, we perform a qualitative analysis across five diverse image classification benchmarks: Food-101 (Bossard et al., 2014b), ImageNet-1K (Russakovsky et al., 2015), CIFAR-100 (Krizhevsky & Hinton, 2009), Caltech-101 (Fei-Fei et al., 2004), and ImageNet-V2 (Recht et al., 2019).

For each dataset, we examine two categories of examples: (1) cases where PI predicts the correct class while the baseline fails, and (2) cases where the baseline succeeds and PI does not. Within each category, we select representative examples showing the largest confidence gaps between the models' top predictions to emphasize where their decisions diverge most significantly.

For each selected image, we visualize the top 5 predicted classes for both models, annotated with their associated confidence scores. Correct classes, when present in the top 5, are highlighted in green. This visualization provides a comparative view of not just prediction correctness, but also the models' relative confidence in those predictions.

Several key trends emerge from this analysis. First, in datasets like Food-101 and Caltech-101, PI demonstrates improved robustness by elevating fine-grained or semantically similar classes in its top predictions, even when the baseline misclassifies with high confidence. Second, in harder benchmarks like ImageNet-V2 and CIFAR-100, while the overall performance margins are smaller, PI still exhibits better calibration—assigning a more conservative confidence score to incorrect guesses, reducing overconfidence. Interestingly, in some baseline-correct examples, PI's top-5 predictions include semantically adjacent classes, suggesting misclassifications that are less egregious than they appear numerically.

The full set of examples for each dataset is presented on separate pages (Figures 2 through 6).

**Dataset: food101**

| Query | PI Top-5 Predictions | Baseline Top-5 Predictions |
|---|---|---|
| | spaghetti bolognese: 0.43
poutine: 0.40
risotto: 0.40
pho: 0.40
paella: 0.40 | fried rice: 0.41
guacamole: 0.41
chicken curry: 0.40
seaweed salad: 0.40
paella: 0.40 |
| | spaghetti bolognese: 0.43
pho: 0.40
ramen: 0.40
risotto: 0.40
pad thai: 0.40 | fried rice: 0.42
spaghetti carbonara: 0.40
caesar salad: 0.40
guacamole: 0.40
spaghetti bolognese: 0.40 |
| | mussels: 0.40
ravioli: 0.38
seaweed salad: 0.38
oysters: 0.38
panna cotta: 0.38 | beet salad: 0.38
escargots: 0.38
mussels: 0.38
oysters: 0.38
spaghetti carbonara: 0.38 |
| | mussels: 0.42
pho: 0.41
panna cotta: 0.40
baby back ribs: 0.40
pulled pork sandwich: 0.40 | grilled salmon: 0.40
guacamole: 0.40
beet salad: 0.40
chicken wings: 0.40
mussels: 0.40 |
| | mussels: 0.42
baby back ribs: 0.40
grilled cheese sandwich: 0.39
grilled salmon: 0.39
panna cotta: 0.39 | grilled salmon: 0.39
mussels: 0.39
spaghetti carbonara: 0.39
guacamole: 0.39
bibimbap: 0.39 |

(a) Cases Where PI Outperformed Baseline

| Query | PI Top-5 Predictions | Baseline Top-5 Predictions |
|---|---|---|
| | risotto: 0.39
spaghetti bolognese: 0.39
fried rice: 0.39
lasagna: 0.39
seaweed salad: 0.38 | fried rice: 0.43
caesar salad: 0.41
seaweed salad: 0.41
cheese plate: 0.40
beet salad: 0.40 |
| | sushi: 0.39
sashimi: 0.38
tuna tartare: 0.38
grilled salmon: 0.38
beef tartare: 0.38 | sashimi: 0.43
sushi: 0.41
beet salad: 0.41
steak: 0.40
takoyaki: 0.40 |
| | takoyaki: 0.37
gyoza: 0.37
bibimbap: 0.37
falafel: 0.37
ceviche: 0.37 | paella: 0.41
sashimi: 0.41
garlic bread: 0.41
seaweed salad: 0.40
carrot cake: 0.40 |
| | caesar salad: 0.38
baklava: 0.38
guacamole: 0.38
foie gras: 0.37
seaweed salad: 0.37 | guacamole: 0.42
breakfast burrito: 0.41
beet salad: 0.40
fried rice: 0.40
grilled cheese sandwich: 0.40 |
| | cheesecake: 0.38
fried rice: 0.38
french toast: 0.37
panna cotta: 0.37
guacamole: 0.37 | guacamole: 0.41
tuna tartare: 0.40
cheese plate: 0.40
caprese salad: 0.40
beef tartare: 0.40 |

(b) Cases Where Baseline Outperformed PI

Figure 2: **Representative examples on the Food-101 dataset comparing predictions from the Perceptual Initialization (PI) model and the baseline.** Each row shows the query image followed by the top-5 predicted classes with associated confidence scores. (a) PI predicts the correct label (in green) while the baseline does not. (b) Baseline predicts the correct label while PI fails.

**Dataset: imagenet1k**

| Query | PI Top-5 Predictions | Baseline Top-5 Predictions |
|---|---|---|
| | zebra: 0.38
impala (antelope): 0.37
cheetah: 0.36
African bush elephant: 0.36
Gila monster: 0.36 | African bush elephant: 0.35
desert grassland whiptail lizard: 0.34
rugby ball: 0.34
African wild dog: 0.34
vulture: 0.34 |
| | zebra: 0.39
impala (antelope): 0.37
Scottish Deerhound: 0.37
cheetah: 0.36
African wild dog: 0.36 | English Springer Spaniel: 0.36
Welsh Springer Spaniel: 0.35
Irish Water Spaniel: 0.35
desert grassland whiptail lizard: 0.35
African bush elephant: 0.35 |
| | king penguin: 0.41
white stork: 0.37
black stork: 0.37
mongoose: 0.37
great egret: 0.36 | Australian Silky Terrier: 0.37
white stork: 0.37
desert grassland whiptail lizard: 0.37
meerkat: 0.37
black stork: 0.37 |
| | snow leopard: 0.40
leopard: 0.38
cheetah: 0.38
tabby cat: 0.38
tiger cat: 0.38 | Gila monster: 0.36
Geoffroy's spider monkey: 0.36
tabby cat: 0.36
Siberian Husky: 0.36
snow leopard: 0.35 |
| | cheetah: 0.39
leopard: 0.38
prairie grouse: 0.37
snow leopard: 0.37
Gila monster: 0.36 | desert grassland whiptail lizard: 0.35
tabby cat: 0.35
Sussex Spaniel: 0.35
peafowl: 0.35
guinea pig: 0.35 |

**(a) Cases Where PI Outperformed Baseline**

| Query | PI Top-5 Predictions | Baseline Top-5 Predictions |
|---|---|---|
| | drum: 0.35
cello: 0.35
violin: 0.35
acoustic guitar: 0.35
ping-pong ball: 0.35 | cello: 0.39
acoustic guitar: 0.38
drum: 0.38
carved pumpkin: 0.37
violin: 0.37 |
| | go-kart: 0.40
messenger bag: 0.40
power drill: 0.39
hair dryer: 0.39
tool kit: 0.39 | chainsaw: 0.44
revolver: 0.42
forklift: 0.42
backpack: 0.42
vacuum cleaner: 0.41 |
| | hot tub: 0.38
cherimoya (custard apple): 0.37
bathtub: 0.37
hamster: 0.37
toucan: 0.37 | bathtub: 0.41
cherimoya (custard apple): 0.39
hamster: 0.39
loggerhead sea turtle: 0.39
dung beetle: 0.39 |
| | hot tub: 0.38
bathtub: 0.37
swim trunks / shorts: 0.36
stingray: 0.35
water jug: 0.35 | bathtub: 0.42
hot tub: 0.40
dough: 0.39
swim trunks / shorts: 0.39
dowitcher: 0.39 |
| | little blue heron: 0.37
pelican: 0.36
spoonbill: 0.36
crane bird: 0.36
black stork: 0.36 | pelican: 0.41
black swan: 0.40
crane bird: 0.38
red-breasted merganser: 0.38
sulphur-crested cockatoo: 0.38 |

**(b) Cases Where Baseline Outperformed PI**

Figure 3: **Representative examples on the ImageNet-1k dataset illustrating qualitative differences between the PI and baseline models.** Each row shows the query image followed by the top-5 predicted classes with associated confidence scores. (a) PI predicts the correct label (in green) while the baseline does not. (b) Baseline predicts the correct label while PI fails.

**Dataset: vtab-cifar100**

| Query | PI Top-5 Predictions | Baseline Top-5 Predictions |
|---|---|---|
| | palm_tree: 0.39
pine_tree: 0.37
rocket: 0.37
cloud: 0.37
willow_tree: 0.36 | sunflower: 0.36
rocket: 0.36
caterpillar: 0.36
snail: 0.36
mountain: 0.35 |
| | squirrel: 0.40
lizard: 0.37
maple_tree: 0.37
oak_tree: 0.37
snail: 0.36 | hamster: 0.37
squirrel: 0.36
kangaroo: 0.36
caterpillar: 0.36
snail: 0.35 |
| | leopard: 0.38
mushroom: 0.37
tiger: 0.36
squirrel: 0.36
cattle: 0.36 | hamster: 0.35
kangaroo: 0.35
turtle: 0.35
leopard: 0.34
squirrel: 0.34 |
| | tiger: 0.37
leopard: 0.36
cattle: 0.34
chimpanzee: 0.34
kangaroo: 0.34 | leopard: 0.33
kangaroo: 0.33
lion: 0.33
caterpillar: 0.33
lobster: 0.33 |
| | palm_tree: 0.39
cloud: 0.37
pine_tree: 0.36
rocket: 0.36
skyscraper: 0.36 | sunflower: 0.36
rocket: 0.36
snail: 0.35
kangaroo: 0.35
mountain: 0.35 |

(a) Cases Where PI Outperformed Baseline

| Query | PI Top-5 Predictions | Baseline Top-5 Predictions |
|---|---|---|
| | orange: 0.37
hamster: 0.37
rabbit: 0.36
bowl: 0.36
pear: 0.36 | woman: 0.41
girl: 0.41
hamster: 0.40
orange: 0.40
poppy: 0.40 |
| | aquarium_fish: 0.37
sunflower: 0.36
orange: 0.35
chimpanzee: 0.35
poppy: 0.35 | sunflower: 0.41
tulip: 0.38
orchid: 0.38
orange: 0.38
aquarium_fish: 0.37 |
| | squirrel: 0.36
tulip: 0.36
chimpanzee: 0.36
snail: 0.36
skunk: 0.36 | tulip: 0.43
sunflower: 0.40
poppy: 0.39
orange: 0.39
caterpillar: 0.39 |
| | poppy: 0.37
sunflower: 0.37
orange: 0.37
tulip: 0.36
mushroom: 0.35 | sunflower: 0.41
tulip: 0.39
orange: 0.38
poppy: 0.37
mushroom: 0.36 |
| | butterfly: 0.37
tulip: 0.36
skunk: 0.35
orange: 0.35
snail: 0.35 | tulip: 0.42
orchid: 0.39
mushroom: 0.38
skunk: 0.37
kangaroo: 0.37 |

(b) Cases Where Baseline Outperformed PI

Figure 4: **Representative examples on the CIFAR-100 dataset illustrating qualitative differences between the PI and baseline models.** Each row shows the query image followed by the top-5 predicted classes with associated confidence scores. (a) PI predicts the correct label (in green) while the baseline does not. (b) Baseline predicts the correct label while PI fails.

**Dataset: vtab-caltech101**

| Query | PI Top-5 Predictions | Baseline Top-5 Predictions |
|-------|---------------------|---------------------------|
| | okapi: 0.38
leopard: 0.35
wild cat: 0.34
brontosaurus: 0.33
panda: 0.33 | body of a cougar cat: 0.35
face of a cougar cat: 0.35
okapi: 0.35
kangaroo: 0.35
wild cat: 0.35 |
| | joshua tree: 0.35
euphonium: 0.34
sea horse: 0.32
bonsai: 0.32
pyramid: 0.31 | kangaroo: 0.32
euphonium: 0.31
bonsai: 0.31
sunflower: 0.31
leopard: 0.31 |
| | joshua tree: 0.35
euphonium: 0.34
sea horse: 0.32
brontosaurus: 0.31
emu: 0.31 | kangaroo: 0.33
bonsai: 0.32
sunflower: 0.32
leopard: 0.32
euphonium: 0.32 |
| | okapi: 0.39
crayfish: 0.37
brontosaurus: 0.37
wild cat: 0.37
stegosaurus: 0.36 | kangaroo: 0.36
rhino: 0.36
emu: 0.36
wild cat: 0.36
ibis: 0.35 |
| | panda: 0.37
okapi: 0.35
dalmatian: 0.35
soccer ball: 0.35
llama: 0.34 | face of a cougar cat: 0.34
wild cat: 0.34
llama: 0.34
soccer ball: 0.33
panda: 0.33 |

**(a) Cases Where PI Outperformed Baseline**

| Query | PI Top-5 Predictions | Baseline Top-5 Predictions |
|-------|---------------------|---------------------------|
| | cannon: 0.37
revolver: 0.37
electric guitar: 0.37
stapler: 0.37
mandolin: 0.37 | revolver: 0.43
motorbike: 0.41
wrench: 0.41
helicopter: 0.40
headphone: 0.40 |
| | wheelchair: 0.36
side of a car: 0.36
motorbike: 0.36
mandolin: 0.35
gramophone: 0.35 | motorbike: 0.43
wheelchair: 0.41
strawberry: 0.40
helicopter: 0.40
flamingo: 0.40 |
| | wheelchair: 0.37
motorbike: 0.36
side of a car: 0.36
lotus: 0.35
sea horse: 0.34 | motorbike: 0.42
wheelchair: 0.40
lotus: 0.38
inline skate: 0.38
platypus: 0.38 |
| | wheelchair: 0.37
side of a car: 0.37
motorbike: 0.36
lotus: 0.36
euphonium: 0.35 | motorbike: 0.42
wheelchair: 0.40
rhino: 0.40
inline skate: 0.39
lotus: 0.39 |
| | wheelchair: 0.39
motorbike: 0.38
side of a car: 0.36
sea horse: 0.36
euphonium: 0.36 | motorbike: 0.44
wheelchair: 0.41
flamingo: 0.41
lobster: 0.41
strawberry: 0.40 |

**(b) Cases Where Baseline Outperformed PI**

Figure 5: **Representative examples on the Caltech-101 dataset illustrating qualitative differences between the PI and baseline models.** Each row shows the query image followed by the top-5 predicted classes with associated confidence scores. (a) PI predicts the correct label (in green) while the baseline does not. (b) Baseline predicts the correct label while PI fails.

**Dataset: imagenetv2**

| Query | PI Top-5 Predictions | Baseline Top-5 Predictions |
|---|---|---|
| | semi-trailer truck: 0.44
tow truck: 0.41
garbage truck: 0.40
recreational vehicle: 0.40
pickup truck: 0.39 | tow truck: 0.39
semi-trailer truck: 0.38
garbage truck: 0.38
jeep: 0.37
pickup truck: 0.37 |
| | impala (antelope): 0.42
African wild dog: 0.38
African bush elephant: 0.38
arabian camel: 0.37
ostrich: 0.37 | red wolf or maned wolf: 0.36
desert grassland whiptail lizard: 0.35
black stork: 0.35
Welsh Springer Spaniel: 0.35
arabian camel: 0.35 |
| | impala (antelope): 0.39
red wolf or maned wolf: 0.35
prairie grouse: 0.35
African wild dog: 0.35
African bush elephant: 0.35 | red wolf or maned wolf: 0.36
flamingo: 0.35
desert grassland whiptail lizard: 0.34
oystercatcher: 0.34
impala (antelope): 0.34 |
| | impala (antelope): 0.41
arabian camel: 0.36
African wild dog: 0.35
wild boar: 0.35
ostrich: 0.35 | red wolf or maned wolf: 0.37
desert grassland whiptail lizard: 0.36
Siberian Husky: 0.36
black stork: 0.36
Welsh Springer Spaniel: 0.36 |
| | impala (antelope): 0.41
prairie grouse: 0.37
African wild dog: 0.36
mongoose: 0.36
desert grassland whiptail lizard: 0.36 | desert grassland whiptail lizard: 0.37
red wolf or maned wolf: 0.37
crane bird: 0.36
impala (antelope): 0.36
kite (bird of prey): 0.36 |

(a) Cases Where PI Outperformed Baseline

| Query | PI Top-5 Predictions | Baseline Top-5 Predictions |
|---|---|---|
| | acorn squash: 0.39
zebra: 0.38
monarch butterfly: 0.38
carved pumpkin: 0.38
spiral or coil: 0.38 | monarch butterfly: 0.43
baguette: 0.39
spiral or coil: 0.39
centipede: 0.38
spaghetti squash: 0.38 |
| | Cardigan Welsh Corgi: 0.35
hockey puck: 0.35
wok: 0.35
Entlebucher Sennenhund: 0.35
Afghan Hound: 0.35 | pool table: 0.40
trifle: 0.39
Tibetan Mastiff: 0.39
hockey puck: 0.38
Bullmastiff: 0.38 |
| | barbershop: 0.36
restaurant: 0.36
bookstore: 0.35
movie theater: 0.35
menu: 0.35 | restaurant: 0.39
shoji screen / room divider: 0.39
movie theater: 0.39
gymnastic horizontal bar: 0.39
barbershop: 0.39 |
| | beer bottle: 0.39
soda bottle: 0.39
totem pole: 0.39
Gila monster: 0.39
smooth green snake: 0.38 | product packet / packaging: 0.43
ruler measuring stick: 0.41
military hat (bearskin or shako): 0.41
revolver: 0.41
sidewinder rattlesnake: 0.41 |
| | sewing machine: 0.40
padlock: 0.40
ladle: 0.39
spindle: 0.39
magnetic compass: 0.39 | revolver: 0.43
fishing casting reel: 0.43
electrical switch: 0.42
sewing machine: 0.42
gas mask or respirator: 0.42 |

(b) Cases Where Baseline Outperformed PI

Figure 6: **Representative examples on the ImageNet-v2 dataset illustrating qualitative differences between the PI and baseline models.** Each row shows the query image followed by the top-5 predicted classes with associated confidence scores. (a) PI predicts the correct label (in green) while the baseline does not. (b) Baseline predicts the correct label while PI fails.

## E.2 Retrieval Tasks

To assess the behavioral differences between Perceptual Initialization (PI) and the baseline in cross-modal retrieval, we present qualitative comparisons for both text-to-image and image-to-text retrieval on a subset of 5000 images from MS-COCO (Lin et al., 2015), which contains diverse natural photos. For each modality, we present two types of examples: (1) cases where PI retrieves semantically more aligned results than the baseline, and (2) failure cases where PI performs worse than the baseline.

In text-to-image retrieval, PI consistently ranks semantically aligned images higher than the baseline, especially in challenging scenes involving object co-occurrence or nuanced spatial relations (e.g., "two elephants walk in the grass together by trees" or "a woman rolling down a sand dune with a red frisbee"). Even in failure cases, PI's top-5 often contain visually coherent distractors, reflecting better alignment despite slight ranking losses. Conversely, baseline retrieval failures tend to retrieve visually dissimilar or irrelevant content, particularly in cluttered or complex scenes.

In image-to-text retrieval, PI excels at capturing fine-grained semantics, such as actions or contextual modifiers (e.g., "a surfer in a wetsuit riding a wave" or "man and woman with luggage near a doorway on a city street"), which the baseline frequently overlooks. When PI fails, the mismatches are often

subtle, involving minor contextual confusions. Overall, PI demonstrates improved grounding and compositional understanding, especially under ambiguous or high-entropy query scenarios. Visual comparisons across representative examples Figures 7 and 10 highlight PI's stronger semantic fidelity and retrieval confidence under both modalities.

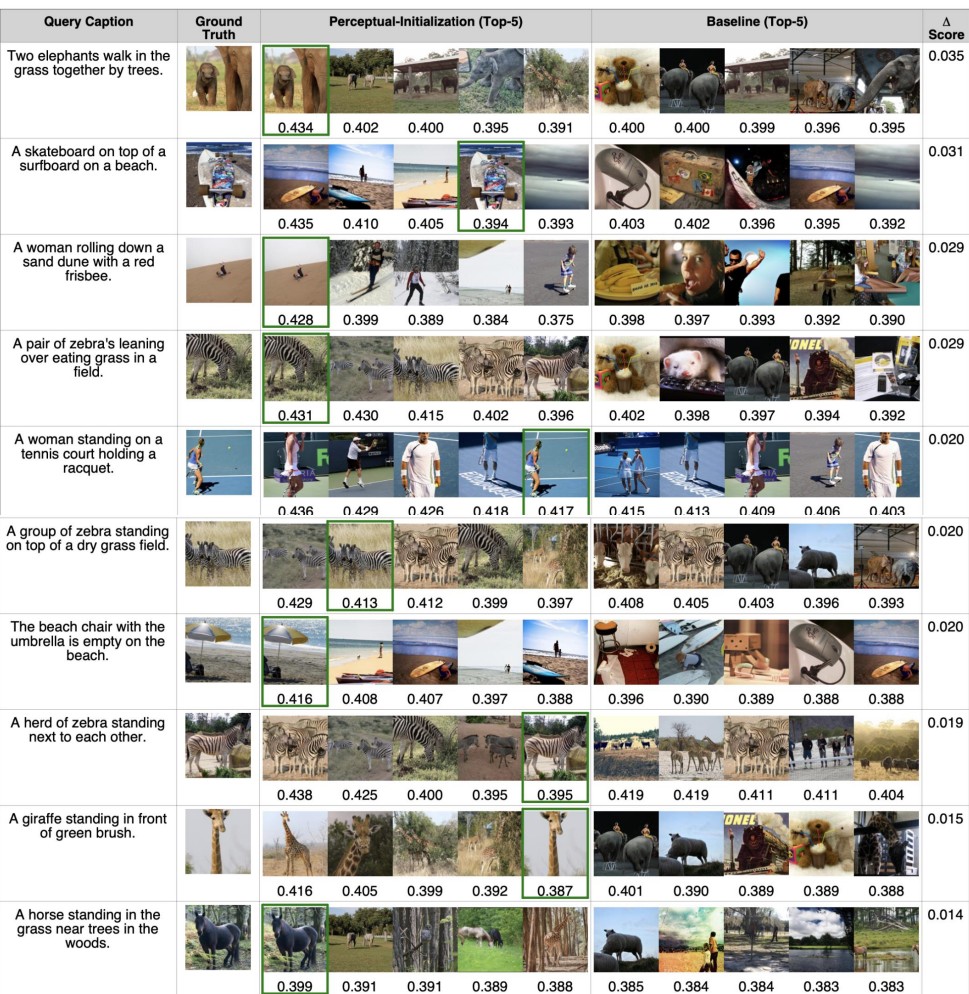

Figure 7: **Representative examples where Perceptual Initialization (PI) outperforms the Baseline on text-to-image retrieval.** Each row shows a query caption, the ground-truth image, and top-5 retrieved images from PI and Baseline models with similarity scores. Green boxes indicate correct retrievals. Overall PI more consistently retrieves semantically accurate and visually aligned results.

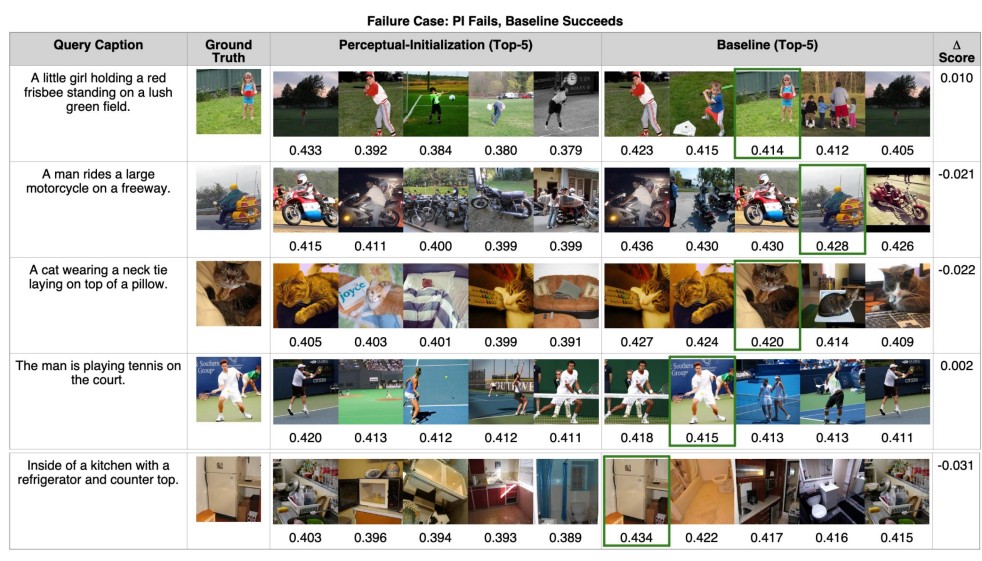

Figure 8: **Failure cases where the Baseline model correctly retrieves the ground-truth image, while Perceptual Initialization (PI) fails.** Although it may fail to retrieve the exact ground-truth, PI frequently presents visually coherent alternatives, demonstrating its semantic sensitivity even when strict retrieval rankings are not met.

| Image | Ground Truth | Perceptual-Initialization (Top-5) | Baseline (Top-5) | Δ Score |
|---|---|---|---|---|
|  | **A train on a track traveling through a countryside.** Commuter train on tracks in rural area on clear day. A high speed passenger train that is going down the track. A long train going down the train track. A train is moving on tracks in an open field. | A purple train traveling down tracks near a platform. (0.436) A train is traveling along train tracks under a signal. (0.425) A blue train going down the train tracks. (0.421) Train pulling up on the tracks next to a stationary double decker train (0.421) **A train on a track traveling through a countryside. (0.420)** | Train pulling up on the tracks next to a stationary double decker train (0.424) A long yellow and red train traveling down tracks. (0.421) A red train traveling down the tracks past grass and trees. (0.416) A red and black train is coming down the tracks (0.416) A couple of large long trains on a track. (0.415) | 0.012 |
|  | A young black bear moves across a grassy field. A large black bear walking across a lush green field. A bear that is walking in a field. **Small black bear in the middle of a flowered field.** **A black bear walking around in the grass during the day.** | **A black bear walking around in the grass during the day. (0.427)** A black bear that is walking on a rock pathway. (0.394) **Small black bear in the middle of a flowered field. (0.392)** A baby elephant following its parent through a field. (0.390) Several sheep in a grassy field near a crow in flight. (0.389) | Several sheep in a grassy field near a crow in flight. (0.416) A black bear that is walking on a rock pathway. (0.409) A black and white cow standing in a field. (0.405) A black and white cow is looking through a fence. (0.405) A black and white dog with a frisbee in its mouth. (0.404) | 0.011 |
|  | The man is taking a photo with his cel phone. A man getting ready to take a picture in a field. **A man in a field takes a picture with his phone.** A man in a field holding up his cell phone. A man is taking a picture with a cell phone. | A man sits on a pile of logs beside his horse. (0.410) **A man in a field takes a picture with his phone. (0.407)** A woman is sitting at a park bench holding her purse and with her other hand is pointing her finger up next to a bronze statue of a man. (0.400) A man in blue shirt feeding a giraffe behind a fence. (0.399) Man in black shirt holding out his hand to a cow. (0.397) | Woman taking a selfie with a giraffe in an enclosure. (0.401) Woman and man feeding giraffes behind a fence outside. (0.401) A flock of sheep with a young boy holding one (0.400) A woman is sitting at a park bench holding her purse and with her other hand is pointing her finger up next to a bronze statue of a man. (0.398) A bearded shirtless man cuddling with a teddy bear. (0.396) | 0.008 |
|  | A group of zebra standing on top of a dry grass field. **Some very cute zebras in a field of tall grass.** A group of zebras that are standing in the grass. Four zebras standing in the tall dry grass **Zebras standing in tall dry grass look at the tourists** | **Some very cute zebras in a field of tall grass. (0.441)** **Zebras standing in tall dry grass look at the tourists (0.440)** A bunch of zebras are standing in a field (0.427) Two zebras are feeding on the grass by themselves. (0.425) A giraffe and a zebra are standing in a field (0.422) | A giraffe and a zebra are standing in a field (0.433) A herd of zebras standing in a dirt field. (0.428) A bunch of zebras are standing in a field (0.423) A giraffe and a zebra eating together in a park. (0.418) Some gazelle and a zebra standing in a field. (0.414) | 0.008 |
|  | A stop sign at the intersection of lyndon ave and south. A road and stop sign at an intersection by a large tree. A stop sign in foreground of a tree in fall A stop sign on the corner of lynden avenue and south street. **A close up of a stop sign under a street sign** | A four-way stop sign is at the corner of delta and bridge street. (0.434) Four way stop sign at street intersection and two street signs above (0.423) A street sign at an intersection of library way and madison avenue (0.418) **A close up of a stop sign under a street sign (0.417)** The cars has stopped at the red stop sign (0.413) | A closeup of a street sign for "main street" with a sign for the wisconsin state fair (0.427) Four way stop sign at street intersection and two street signs above (0.421) A street sign above a speed limit sign on a rural street. (0.421) A one-way sign at library way and madison ave. (0.421) A fire hydrant and a street sign are on the side of a street. (0.421) | 0.007 |
|  | Two people outside of a stone building near a red fire hydrant. There are two people standing on the side of a street. A woman is pulling luggage on to a sidewalk near a fire hydrant **Man and woman with luggage near a doorway on a city street.** A city scene of a sidewalk with a red fire hydrant on a sidewalk next to an atm | **Man and woman with luggage near a doorway on a city street. (0.409)** Two people sitting at a bench on a city street (0.402) A boy rides a skateboard next to men walking down a street. (0.399) A building standing in front of a street with a crossing section (0.398) Three stop lights and a woman walking across a street. (0.397) | A motorcycle is pictured outside of a building with a man walking away from it. (0.402) A person and some cones on a city street. (0.401) A group of people walking past the front of a store. (0.399) A corner of a street next to a building (0.397) Two men wearing back packs walking through a park in the city. (0.396) | 0.006 |
|  | **A blue and yellow train pulling up to a station on a sunny day.** A bluish train car pulls into a station. A passenger train on a track next to a station. **A blue and yellow train in the train station.** A commuter train pulling into the train station | **A blue and yellow train pulling up to a station on a sunny day. (0.429)** **A blue and yellow train in the train station. (0.401)** A blue train going down the train tracks. (0.396) A train with bright yellow engine on tracks beside tall leafy trees. (0.392) A black train and orange train cars on tracks. (0.391) | A train with bright yellow engine on tracks beside tall leafy trees. (0.423) A black train and orange train cars on tracks. (0.422) A yellow and black train is coming down the tracks (0.421) Train pulling up on the tracks next to a stationary double decker train (0.417) A large blue passenger train pulling into a train station. (0.412) | 0.005 |
|  | A young man riding a skateboard up a black ramp. A skateboarding boy is about to go onto the red skate bar. A skateboarder starting a jump on a homemade ramp. A person standing on a skateboard and performing a stunt on a platform in the street. **A man is riding a skateboard up a ramp on a street in front of a truck.** | A worker driving a cart pulling a trailer loaded with cargo. (0.411) A man holding glass near a pick up truck on the street. (0.411) **A man is riding a skateboard up a ramp on a street in front of a truck. (0.407)** Two boys moving along outside during the day, one of them has a skateboard. (0.404) A man is trying to pull off a skateboarding trick on his ramp. (0.398) | A couple of men are loading a truck with glass (0.407) Two women eat chili dogs on a city sidewalk. (0.404) A woman riding a bike down the street (0.403) Two boys getting ready to go down the skateboard ramp on their skateboards. (0.402) A man and woman loading a surfboard on a motorcycle outside with other riders nearby (0.402) | 0.004 |
|  | A one way sign pointing to the left; the sky is blue in the background. A group of different road signs, one of which is a one way only sign. A one way sign mounted to the side of a pole. A close up of a one way sign on a pole. **A pole with a one way street sign as well as a few others** | Four way stop sign at street intersection and two street signs above (0.436) A close up a street pole with a homemade street sign. (0.425) A one way, left turn only, straight only, no right turn, street sign and traffic light. (0.425) **A pole with a one way street sign as well as a few others (0.422)** A street sign at an intersection of library way and madison avenue (0.422) | A pole holding the street sign for queen street is decorated with a painting of a queen. (0.432) Street light with street signs that restrict traffic. (0.430) Someone walking down the street and street lights and a one way street sign (0.430) Four way stop sign at street intersection and two street signs above (0.429) A closeup of a street sign for "main street" with a sign for the wisconsin state fair (0.429) | 0.004 |
|  | A man riding a wave on top of a surfboard. A male surfer dressed in black riding a white surfboard. The man is riding the waves on his surf board. A man in a wet suit surfs a wave. **A surfer in a wetsuit riding a wave** | A person in a wetsuit riding a wave on a surfboard. (0.433) A surfer in a wetsuit rides on a wave. (0.430) **A surfer in a wetsuit riding a wave (0.430)** A man on a surfboard riding the waves in the ocean (0.422) Two people riding surf boards on a wave (0.421) | A man on a surfboard surfs through the waves of a windy coast. (0.429) A man on a surfboard riding a wave (0.428) A man surfing on his surf board against the waves (0.428) A young male riding an ocean wave on a surfboard (0.428) A man riding a wave on a surfboard in the ocean. (0.427) | 0.004 |

Figure 9: **Representative examples where Perceptual Initialization (PI) outperforms the Baseline on image-to-text retrieval.** Each row shows a query image, the ground-truth caption, and the top-5 retrieved captions from PI and Baseline models along with similarity scores. Bold text indicates correct retrievals. PI frequently retrieves captions that are more descriptive or semantically grounded, often outperforming the baseline in nuanced language alignment.

**Failure Case: PI Fails, Baseline Succeeds**

| Image | Ground Truth | Perceptual-Initialization (Top-5) | Baseline (Top-5) | Δ Score |
|---|---|---|---|---|
|  | A piece of partially-eaten cake sits on a paper plate. 
 A desert topped with gummy bears sits on a yellow and orange plate. 
 a close up of a plate of food with a fork 
 **A slice of chocolate cake with dark chocolate glaze.** 
 **A slice of chocolate cake with dark chocolate icing.** | A large piece of blueberry cake on a plate. (0.409) 
 A plate with some meat sitting on top of it (0.405) 
 Picture of food in restaurant - grilled meat on white plate with orange drink and side dish (0.400) 
 A plate filled with french toast sitting next to a drink. (0.398) 
 Two slices of cake sitting on top of a white plate. (0.398) | A large piece of blueberry cake on a plate. (0.440) 
 **A slice of chocolate cake with dark chocolate glaze. (0.439)** 
 **A slice of chocolate cake with dark chocolate icing. (0.432)** 
 A chocolate and vanilla birthday cake sitting on a white table. (0.432) 
 A piece of white cake with a topping sits on a white plate on a table. (0.426) | -0.031 |
|  | **A shirtless man playing tennis on a blue court.** 
 an image of a shirtless man hitting tennis racket 
 A man is standing on a court with a tennis racket. 
 A man on a court swinging a tennis racket. 
 This man is playing tennis without a shirt. | Man on a tennis court holding a racket and playing badminton (0.406) 
 A man jumping up to hit a tennis ball with a racket. (0.381) 
 The woman are playing tennis on the court. (0.381) 
 The man is playing tennis on the court. (0.380) 
 A woman swinging a tennis racket during a match. (0.380) | **A shirtless man playing tennis on a blue court. (0.424)** 
 A woman tennis player playing tennis in the stadium. (0.421) 
 Women on a tennis court playing a doubles match. (0.421) 
 A woman is holding a tennis racket on a court (0.418) 
 Female tennis player playing on a tennis court (0.416) | -0.018 |
|  | **A man walking behind a row of parked cars holding an umbrella.** 
 It is a good thing he has his umbrella while walking in the parking lot. 
 **A solitary man walks through a crowded parking lot with his striped umbrella.** 
 A man holding a white and black umbrella in a large parking lot. 
 A man with an umbrella walks along a row of cars. | A large boat and two smaller boats in a row. (0.420) 
 A back ally neighborhood with motor bikes in a row (0.404) 
 A number of people on a beach holding surf boards (0.403) 
 There are many people and umbrellas on the beach (0.402) 
 The boat includes several rows of orange chaird. (0.401) | **A man walking behind a row of parked cars holding an umbrella. (0.413)** 
 A row of mopeds parked outside of a building. (0.412) 
 There are many people and umbrellas on the beach (0.412) 
 **A solitary man walks through a crowded parking lot with his striped umbrella. (0.411)** 
 A row of motorcycles parked in front of a building. (0.409) | 0.007 |
|  | The striped cat is sitting on top of the car. 
 a cat sitting on a car engine with the hood up 
 someone opened a hood on a car and he cat jumped up on the edge 
 Black cat sitting on the engine of a black car. 
 **A gray cat is walking next to a truck.** | A dog on a leash sniffing at a door. (0.406) 
 A man holding a black umbrella walks near a man who has two dogs on leashes down a park path. (0.402) 
 A cat is walking across the dash of a car (0.400) 
 A little dog on a leash is sniffing at a door. (0.396) 
 A dog on a leash standing at a doorway. (0.396) | **A gray cat is walking next to a truck. (0.432)** 
 A black and white cow stands in an enclosure. (0.416) 
 A cat standing next to an open box with pizza in it. (0.413) 
 A cat stands next to an open pizza box. (0.413) 
 A cat watches as its owner uses the laptop. (0.413) | -0.026 |
|  | a couple of people that are surfing in some water 
 Surfers on surfboards are riding a wave together. 
 Several people ride surfboards in the ocean waves. 
 **Surfers braves the waves on the choppy blue ocean.** 
 The people are surfing the ways on the water. | A man on a surfboard surfs through the waves of a windy coast. (0.424) 
 A man on a surfboard, surfing in the ocean. (0.412) 
 A person that is surfboarding through the waves in the ocean. (0.411) 
 A man surfing waves on his surf board (0.411) 
 A man surfing on his surf board against the waves (0.411) | A person riding a surf board on a wave (0.424) 
 A person riding a surf board on a wave (0.424) 
 A man on a surfboard surfs through the waves of a windy coast. (0.419) 
 **Surfers braves the waves on the choppy blue ocean. (0.416)** 
 A man surfing on his surf board against the waves (0.416) | -0.000 |

Figure 10: **Failure cases where Perceptual Initialization (PI) fails but the Baseline succeeds on image-to-text retrieval.** In several cases, PI retrieves captions that are visually relevant but semantically offset, suggesting opportunities for further improving alignment in edge cases.