# OpenReview forum: "Beginning with You: Perceptual-Initialization Improves Vision-Language Representation and Alignment"
_ICLR.cc/2026/Conference — ICLR 2026 Conference Desk Rejected Submission_

### Official Review · Reviewer_rYhZ · 2025-10-25

**Soundness:** 4
**Presentation:** 3
**Contribution:** 3
**Rating:** 6
**Confidence:** 5

**Summary:**

This paper introduces Perceptual-Initialization (PI), which initializes the CLIP vision encoder using human perceptual similarity data (NIGHTS triplets) before standard large-scale image-text contrastive training on YFCC15M.
Compared with random initialization and with post-hoc perceptual fine-tuning, the proposed method yields consistent zero-shot gains across 29 classification and 2 retrieval benchmarks. The authors argue that embedding human perceptual priors at the start of training leads to faster convergence and more human-aligned representations.

**Strengths:**

1. Novel use of human perceptual priors as initialization rather than alignment fine-tuning.
2. Comprehensive evaluation over diverse datasets shows consistent positive gains.
3. Very low additional compute cost.
4. Clear comparison showing that late perceptual fine-tuning disrupts alignment and opens new direction for human or brain aligned pretraining.

**Weaknesses:**

1. No experiments using random or pseudo perceptual triplets to isolate the contribution of human perceptual structure.
2. The approach is validated only on NIGHTS; applicability to richer datasets remains untested.
3. No probing or visualization is provided to show how perceptual initialization changes internal feature space or similarity structure compared to the baseline.

**Questions:**

1. Could the authors analyze which visual attributes benefit most from perceptual initialization (e.g., texture vs. shape bias)?
2. Does PI primarily affect the early layers or propagate to higher-level semantics during contrastive training?
3. How much perceptual data is necessary—does performance saturate after a certain fraction of NIGHTS triplets?
4. Could PI be combined with supervised or robust-CLIP initializations, or would they interfere?

---

### Official Review · Reviewer_NrFX · 2025-11-01

**Soundness:** 4
**Presentation:** 3
**Contribution:** 3
**Rating:** 6
**Confidence:** 4

**Summary:**

This paper presents a two stage training paradigm  for VLMs like CLIP arguing the benefits of perceptual initialization (PI) over random initialization. Further it argues that incorporating PI in initialization phase more advantageous than post-hoc finetuning. The main contribution is demonstrating that this early-stage alignment provides a stronger foundation for general-purpose VLM intelligence. PI models show significant zero-shot performance improvements, without any task-specific fine-tuning, across a comprehensive suite of 29 classification and two retrieval benchmarks. PI approach consistently outperforms a randomly-initialized baseline, and a direct comparison shows that post-hoc perceptual fine-tuning is catastrophic to V-L alignment.

**Strengths:**

**originality**: Lveraging supervised human behavioural data as a foundational inductive bias in the model intialization is a novel idea that opens a new research direction. The works provided a provides a structured solution that converts often ignored variance of random initialiation into a principled prior.

**Significance**:  PI paradigm is the core strength of the paper. It uses the supervised human perceptual data to initialize a VLM parameter prior to large scale pretaining, provide a potent to human aligned inductive bias right from time t=0.

**Quality**: The provided results empirically validate the PI hypothesis, having  consistent performance gains, outperforming 23/29 classification tasks. Further, it shows how post-hoc finetuninig leads to catastrophic forgetting.

**writing**: The argument for PI is presented logically, starting from the known "path-dependency" of deep networks and the variance of random seeds, making the motivation for a principled initialization intuitive.

**Weaknesses:**

**Limited scope of the prior**: Only the vision encoder is initialized with PI and the text encoder is still randomly initialized and trained from scratch. What is the reason for this choice for the experiments?
CLIP like model operates on the shared latent space of vision and text modalities. The paper could be strenthened by exploring complementary intialization of text encoder, to see if such complete model with PI initialization provides synergistic benefits.

**Perceptual Loss**:  The core of the PI benefits lays in the perceptual loss function which is derived from a previous works. There's no/lack of evidence/interpretation (apart from the final results) provided on how does this loss function work/fail in the assumed context:  pretraining vs post-hoc finetuning.

**Mechanistic Analysis**: While efficacy id proven, the paper does not delve into why the inductive bias remains so effective after 32 epochs, where the post-hoc finetuning fails.  This theoretical insight is critical to see the compatibility of leveraging this idea to different models or scenarios. Many of the questions in **Questions** section could not answered from the given content of the paper.

**Limited evaluation**: current training uses 15M image-text pairs, while this is substantial, SOTA VLMs often trained on hunderds of millions or biilions of pairs. Will the proportional gains from PI would persist, diminsh or grow continuously (Though limited scling law provided in the paper). In failure cases, how PI should be addressed?

**Questions:**

- Do the PI weights remain closer to the perceptiual optimum throughout training?
- How does the learned Image-Text alignment module interact differently with the PI-derived features versus the baseline features? How does the shared representation space differ?
- Have any preliminary experiments been conducted to determine the minimal amount of human perceptual data required in Stage 1 to achieve a statistically significant positive gain?
- Why/How does this loss function work?
- Can the authors analyze the evolution of the logit scaling parameter ($\tau$) in Stage 2?
- For failure cases, should the perceptual prior be "re-anchored" at intermediate stages, or perhaps weakened by introducing a temperature parameter to the perceptual loss?

---

### Official Review · Reviewer_uwto · 2025-11-02

**Soundness:** 3
**Presentation:** 4
**Contribution:** 4
**Rating:** 8
**Confidence:** 3

**Summary:**

This paper introduces a new visual representation learning scheme called Perceptual-Initialization, which trains the visual encoder to match human preference before the contrastive learning. Specifically, the human preference alignment is achieved using a triplet contrastive loss on the NIGHT dataset and the resultant model weights are used as the initialization of the formal contrastive learning. PI achieves zero-shot performance improvements on a variety of image classification and retrieval benchmarks compared to the baseline CLIP.

**Strengths:**

- The proposed method is novel, simple yet effective. The promising results of the paper can encourage following researches exploring other initialization strategies.
- Results in zero-shot image classification and retrieval tasks demonstrates that PI scales as the data volume increases, indicating the method's potential in large-scale training.
- The paper is well organized and nicely presented. The ending section points out remaining challenges faithfully and offers valuable insights, strengthening its contribution to the field.

**Weaknesses:**

- The proposed method limits its scope for the initialization of CLIP type model, despite that the human preference alignment is independent to the text encoder. The author could add experiments on other visual backbones such as vanilla ViTs to fully explore the potential of the method.

**Questions:**

- As mentioned in the weakness part, I'm wondering if PI could also benefit other types of visual pretraining?
- Additionally, does the model trained using PI demonstrates stronger transferability compared to normal training?

---

### Author Response · Authors · 2025-12-03
**Author Response to Reviews Part I**

We thank all reviewers for their thoughtful and valuable feedback.

We are encouraged that reviewers recognize the novelty and effectiveness of Perceptual-Initialization (uwto, NrFX, rYhZ), its potential to open new research directions (uwto, NrFX), and our comprehensive evaluation showing consistent improvements across benchmarks. We appreciate the acknowledgment that early-stage PI provides stronger benefits than post-hoc fine-tuning (NrFX, rYhZ).

The **main concerns** are:

(1) Scope and generality beyond CLIP vit B32 (uwto, NrFX).

(2) Mechanistic understanding of why PI helps while post-hoc perceptual fine-tuning fails, and how the perceptual prior persists during CLIP training (NrFX, rYhZ).

(3) Evaluation and controls, including alternative perceptual datasets, scaling to larger models/data, and robustness (NrFX, rYhZ).

(4) Possible extensions to text encoders and additional perceptual datasets (NrFX, rYhZ).

We ran several additional experiments and analyses to address these points:

**Mechanistic understanding: Persistence of perceptual structure**
To address the question of why the inductive bias remains effective after 32 epochs (NrFX, rYhZ), we computed Centered Kernel Alignment (CKA) [1]  between Stage-1 perceptual representations and Stage-2 representations over training. Results show that the PI model starts at CKA 0.663 and gradually decreases to 0.491 by the end of training, whereas the baseline starts much lower at 0.5598 and ends at 0.486. This confirms that PI shapes the optimization basin from $t=0$, allowing the model to preserve the intended perceptual structure throughout pretraining. Weight-spectrum analyses (SVD [2] across 97 layers) further show that PI induces more organized weights (13.8% lower condition numbers) particularly in visual attention layers (spectrum gap 0.25).

**Complementarity with downstream fine-tuning (vs. post-hoc failure)**
Reviewers (NrFX) noted the failure of post-hoc perceptual fine-tuning. We clarify that PI is a foundation that supports and can be extended to downstream adaptation. We tested WiSE-FineTune [3] on top of PI-initialized models. Unlike the failure mode of post-hoc tuning, PI combined with WiSE-FT achieves the best overall performance, outperforming the strong WiSE-FT baseline by +2.8 pp (CIFAR-10), +5.3 pp (CIFAR-100), +0.2 pp (ImageNet-A), and +0.7 pp (ImageNet-R). This directly addresses the concern by showing that while late injection of perceptual data disrupts alignment, early injection (PI) is a complementary approach that creates a fundamentally stronger base model.

**How many human comparisons are needed**
We subsample NIGHTS triplets at 50% and repeat the PI stage. Using only half the human perceptual data, the model outperforms the baseline on several benchmarks. Notable improvements include STL-10 (+9.4 pp), Food-101 (+3.3 pp), DTD texture classification (+3.1 pp), ImageNet-O out-of-distribution detection (+2.6 pp), GTSRB traffic sign recognition (+2.0 pp),  ImageNet-R (+1.9 pp), ObjectNet (+1.8 pp), ImageNet-V2 (+1.0 pp), and CLEVR-Count (+1.3 pp). These results indicate that perceptual initialization remains effective with substantially reduced human perceptual data on certain tasks. We will include results using a 75% subsample of the NIGHTS dataset in the camera-ready version.

**How the perceptual loss (Eq. 2) works and its robustness**
Addressing questions regarding the loss function mechanism (NrFX): The margin $m$ in Eq. 2 enforces a cardinal constraint $d(neg, anchor) \ge d(pos, anchor) + m$, which is stronger than a simple ordinal check. We performed an ablation on Stage 1 with $m \in \{0.01, 0.05, 0.1\}$. Despite varying initial loss magnitudes, all settings converged to essentially the same validation accuracy on NIGHTS (71.0–71.6%). This indicates that the loss effectively encodes the core ordinal relations regardless of the precise margin, and this learned geometry is what persists to guide the CLIP pretraining.

---

### Author Response · Authors · 2025-12-03
**Author Response to Reviews Part II**

**Scaling across architectures**
Beyond CLIP ViT B32, we trained larger and alternative backbones with PI under the same two-stage paradigm. With PI, CLIP ViT-L/14 shows gains of +2.5/+3.2 pp (acc@1/5) on CIFAR-10 and +0.4/+0.5 pp on CIFAR-100 after 8 epochs (around 120M samples). A CLIP RN50 with PI shows +11.6/+5.2 pp on CIFAR-10 and +2.5/+4.2 pp on CIFAR-100 after 15 epochs (around 225M samples). These results indicate that PI’s advantages persist for both transformer and convolutional architectures and at larger effective data scales. Consistent with the paper, PI also increases the scaling-law exponent $\beta$ for classification (Figs. 3–4), indicating faster error decay per byte of data.

While we have addressed the core concerns raised by the reviewers, we acknowledge that several interesting questions remain for future work.These include: exploring complementary initialization strategies for the text encoder, testing PI with alternative perceptual datasets beyond NIGHTS and alternative perceptual architectures beyond CLIP, like DINO, and validating scalability at the hundred-million to billion-scale data regimes. We believe these directions will further strengthen the PI paradigm and look forward to exploring them in future research.

References:

[1] Simon Kornblith, Mohammad Norouzi, Honglak Lee, and Geoffrey Hinton.Similarity of neural network representations revisited.In Proceedings of the 36th International Conference on Machine Learning, volume 97, pp.  3519–3529. PMLR, 09–15 Jun 2019

[2] Martin, C. H., & Mahoney, M. W. (2018). "Implicit Self-Regularization in Deep Neural Networks: Evidence from Random Matrix Theory and Implications for Learning." arXiv preprint arXiv:1810.01075.

[3] Wortsman, M., Ilharco, G., Kim, J. W., Li, M., Kornblith, S., Roelofs, R., Lopes, R. G., Hajishirzi, H., Farhadi, A., Namkoong, H., & Schmidt, L. (2022). Robust fine-tuning of zero-shot models. Proceedings of the IEEE/CVF Conference on Computer Vision and Pattern Recognition (CVPR).

---

### Note · Program_Chairs · 2026-01-17
**Submission Desk Rejected by Program Chairs**

The following references in this submission do not refer to real documents and/or have major errors in bibliographic information:

 Y. Liu, X. Zhang, Y. Wang, and Q. Wang. A deep learning approach for predicting odor perception from molecular structure. In 2022 44th Annual International Conference of the IEEE Engineering in Medicine & Biology Society (EMBC), pp. 1433-1436. IEEE, 2022. doi: 10.1109/EMBC48229. 2022.9720238. URL https://ieeexplore.ieee.org/document/97202